# A local thermal non-equilibrium model for Rain-on-Snow events

Thomas Heinze[1]

[1]Institute of Geology, Mineralogy and Geophysics, Ruhr-University Bochum, Germany

**Correspondence:** Thomas Heinze (thomas.heinze@rub.de)

**Abstract.** Liquid water movement through a snowpack, e.g. during rain-on-snow events or meltwater infiltration, is an essential process to understand runoff generation, flash floods, and snow avalanches. From a physical point of view, water infiltration into snow is a strongly coupled thermo-hydraulic problem with a thermal non-equilibrium between phases because the infiltrating water can be substantially warmer than the snowpack. Contrary to water infiltration into a frozen soil, the solid volume fraction is highly dynamic due to melting of snow and (re-)freezing of water. This work presents the first true multi-phase local thermal non-equilibrium model with variable volume fractions of all involved phases including the snowpack as solid porous matrix. While the possible value range of hydraulic, geometrical, and thermal parameters within a snowpack can be highly variable, the developed model is subsequently used to systematically study the effects of environmental conditions and parameters on the spatial distribution of melting and freezing within the snowpack. The model can be used to identify the formation of new ice layers due to refreezing as well as layers of enhanced melting.

## 1  Introduction

Rain-on-snow (ROS) events are prominent examples of potentially hazardous events controlled by the thermal state of the system which can occur at almost any rainfall intensity (Baselt and Heinze, 2021). Severe floods and mud flows can be observed when warm storm systems release rain on a snow cover because the amount of surface water run-off due to ROS events is increased and accelerated in comparison to natural snow melt under non-rain conditions (Singh et al., 1997). The intruding rainwater can warm and (partly) melt the snowpack changing the liquid-water content as well as the structure of the porous snow matrix (Pfeffer et al., 1990). Prolonged rainfall can wet a snow cover completely or even melt it entirely augmenting runoff (Wei and Gao, 1992), especially in combination with air temperatures above the melting point. Besides flooding, ROS events can also influence the snowpack's stability with respect to avalanches and slush flows due to additional load when rainwater freezes during its passage through the snowpack (e.g. Baggi and Schweizer, 2009). The rainwater freezes within the snowpack when it is cooled by the snow below the freezing point. Then usually the snow characteristics change as ice layers are formed within the snow, also depending on other factors such as snow layering and load. Frozen rainwater inhibits subsequent infiltration by reducing the hydraulic conductivity through blocked pathways (Eiriksson et al., 2013; Pfeffer et al., 1990; Wever et al., 2016).

ROS effects have been observed in Europe as well as in North America (e.g. McCabe et al., 2007; Rössler et al., 2014; Jeong and Sushama, 2018; Li et al., 2019; Juras et al., 2021) and the numbers of ROS events is assumed to increase in the near future

due to climate change as the number of liquid precipitation patterns increase and rain intensity increases (Musselman et al., 2018; Sezen et al., 2020).

In general, the number of models on heat and mass transfers in soils or rocks is substantially larger than for snow. For example, there is an extension of the well-known infiltration model Hydrus-1D solving the Richards equation towards freezing conditions using similarities between soil drying and soil freezing (Spaans and Baker, 1996; Hansson et al., 2004). One-dimensional simulations have also been used to develop advanced numerical schemes, such as splitting mass and heat transport in the soil (Dall'Amico et al., 2011). These models do not consider changes in the soil structure and soil properties that can occur after multiple freeze and thaw cycles (Leuther and Schlüter, 2021). On a regional scale, the extension piFreeze allows catchment scale modeling of soil freezing using the FeFlow modeling software (Langford et al., 2020; Magnin et al., 2017). Further examples and comparison of numerical models is provided within the InterFrost initiative (Grenier et al., 2018). A discontinuous model for thermal non-equilibrium between an ice grain and surrounding liquid water was introduced in Peng et al. (2016) using the Richards equation to describe water infiltration with a sink term to account for the phase change. There are several similarities but also substantial differences in mass and heat transport between snow and frozen soil (Kelleners et al., 2016).

Pioneering works describing water movement within a (partially) saturated snow cover were commonly based on Darcian flow and showcased that capillary effects could be considered minor, hence simplifying the flow in the unsaturated zone (Colbeck, 1972). Early on, a precise description of the energy balance of the snowpack was found crucial for accurately predicting snow melt and runoff (Dunne et al., 1976). Special attention was paid to the parameterization of hydraulic and thermal properties in dependence of saturation and temperature (e.g. Colbeck and Anderson, 1982). This also included phenomena such as grain growth and the decay of ice layers (Colbeck, 1978). Physically, these models were able to simulate nocturnal freezing, diurnal changes (Akan, 1984), percolation of surface melt through the snow (Colbeck and Anderson, 1982) in a multi-phase setting. A review of physics-based models of snow melt with detailed descriptions of boundary conditions is given in Morris (1991). To quantitatively describe heat and mass transport models with varying complexity for freezing and thawing in soil and snow are also presented by Kelleners et al. (2009) and Kelleners (2013). The focus of these models, while being applied to reduced geometrical scenarios, is on field applications including root water uptake and water vapor flow in addition to freezing and melting.

The most common quantitative description of snow and snow hydrology is given by the 1-D SNOWPACK model (e.g. Wever et al., 2014, 2016). The model solves the Richard's equation to include capillary forces on water flow and also utilizes a dual-domain approach to consider preferential flow (Würzer et al., 2017). It further includes snow settling and snow metamorphism. SNOWPACK is also part of the software suite ALPINE3D which couples several modules to simulate mass and heat exchange processes between snow, soil, and atmosphere on three-dimensional alpine surfaces (Lehning et al., 2006). The more general surface modeling platform SURFEX also incorporates detailed snow processes using the 1D snowpack model Crocus (D'Amboise et al., 2017). Crocus solves the 1D Richards equation to simulate water flow during thawing within the snowpack while also accounting for changes in the snow grain morphology (Vionnet et al., 2012). Emphasis has been given to

snow depth and snow density estimations over snow seasons by incorporating interaction with the atmosphere, such as through surface albedo (Viallon-Galinier et al., 2020). An overview on existing snowmelt models is given in Zhou et al. (2021).

All those theoretical and numerical models are based on the Local Thermal Equilibrium (LTE) assumption and therefore of limited use for ROS events, as those ROS events include - at least initially - a local thermal non-equilibrium (LTNE) between the involved phases because the liquid (rain-)water has temperatures above the melting point, while the snow is frozen at temperatures at or below the freezing temperature. A local thermal non-equilibrium model for water infiltration into frozen soil has been presented in Heinze (2021) deriving a multi-phase heat transfer model in combination with a hydraulic unsaturated flow model for comparably warm water infiltrating into an initially frozen soil. A clear occurrence and sustainability of temperature differences between phases has been observed with increasing effects with increasing soil grain size. However, apart from the initial LTNE conditions, the sustainability and possible consequences of LTNE during ROS events remain unknown.

This work investigates under which meteorological and hydraulic/thermal snow conditions LTNE can sustain and how LTNE potentially affects the thermo-hydraulic response of the snow pack to the rain. To address these questions, a one-dimensional LTNE model for a snow pack with infiltrating water with different phase temperatures is developed. Due to the possible phase change between liquid water and snow, the replacement of air by infiltrating water, and the possible melting of the snowpack representing the porous matrix, the described scenario requires substantial modifications and extensions of previous work originally developed for a static porous matrix (Heinze, 2021). The developed model is compared to a LTE model and historic field observations to investigate the models capabilities and performance. Subsequent simulations varying meteorological and thermo-hydraulic parameters investigate which conditions promote LTNE and address the effect of uncertainty in model parameterization.

## 2 Mathematical and numerical ROS model

### 2.1 Water infiltration into snowpack

Water infiltration into the snowpack will be described by the Richard's equation (Richards, 1931) in the head based form (Farthing and Ogden, 2017):

$$c(\psi)\frac{\partial \psi}{\partial t} = \boldsymbol{\nabla} \cdot (K(\psi,\epsilon_i)\boldsymbol{\nabla}(\psi - z)) + M \tag{1}$$

with the matrix potential $\psi$ (m), the specific moisture capacity $c$ (m$^{-1}$), the hydraulic conductivity $K$ (m s$^{-1}$), the volumetric ice content $\epsilon_i$ and possible source or sink $M$ (s$^{-1}$). The z-axis is defined positive in downward direction. The relationship between effective water saturation and the hydraulic head is given by the model of van Genuchten (1980) and the respective parameters $\alpha$ (m$^{-1}$), $n$ (-) and $m = 1 - 1/n$:

$$S_{eff} = \frac{\epsilon_l - \epsilon_{l,res}}{\epsilon_{l,sat} - \epsilon_{l,res}} = (1 + |\alpha\psi|^n)^{-m}. \tag{2}$$

with the saturated $sat$ and residual $res$ liquid $l$ water content. Throughout this work the subscripts $i$ (ice), $l$ (liquid water), and $a$ (air) will be used respectively. Subsequently, the moisture capacity is given by van Genuchten (1980)

$$c(\psi) = \alpha \cdot m \cdot (\epsilon_{l,sat} - \epsilon_{l,res}) \cdot (1-m)^{-1} \cdot S_{eff}^{1/m} \cdot \left(1 - S_{eff}^{1/m}\right)^m. \tag{3}$$

While the Richard's equation and the van Genuchten relationship are usually applied for soil, their applicability to snow has been shown in the past (Kelleners et al., 2016). In general, the Richard's equation using the van Genuchten relationship improved the runoff estimation of meltwater in a multi-layered snowpack compared to simpler approaches (Hirashima et al., 2010; Wever et al., 2014). In experimental studies, strong similarities in water retention curves of sands and snow were observed and the van Genuchten parameters $(\alpha, n)$ have been found to depend on snow properties and therefore to vary over a winter season (Yamaguchi et al., 2010). Subsequently, the van Genuchten parameters were related to grain size and bulk dry density by various formulations (Yamaguchi et al., 2012), but the most common relationships are empirically derived from experiments in dependence of the grain diameter $d$ (m) (Yamaguchi et al., 2010; Daanen and Nieber, 2009). These links of the van Genuchten relationship with snow properties have also been successfully used for snow strength estimations affected by increasing water content above hydraulic barriers (Schlumpf et al., 2024). Known hysteresis of the water retention curves is neglected here for simplicity. Such hysteresis stems from the various pore shapes of snow, which cause different saturation responses to changes of the hydraulic pore pressure during wetting and draining (Leroux and Pomeroy, 2017). Here, the formulas presented in Yamaguchi et al. (2010) will be applied

$$\alpha = 7.3 \, [m^{-1}/mm] \cdot d \, [mm] + 1.9 \, [m^{-1}], \tag{4}$$

$$n = -3.3 \, [mm^{-1}] \cdot d[mm] + 14.4. \tag{5}$$

For soils, if ice grains block the fluid pathway in a porous media, the hydraulic conductivity is reduced. This can be represented by an exponential impedance factor $\Omega$ (-), which has been set to 7 in previous studies (Hansson et al., 2004; Dall'Amico et al., 2011; Peng et al., 2016). The hydraulic conductivity in dependence of liquid water saturation and volumetric ice content $\epsilon_i$ is then given as

$$K(\psi, \epsilon_i) = K_{sat} \sqrt{S_{eff}} \left(1 - (1 - S_{eff})^{1/m}\right)^2 \cdot 10^{-\Omega \epsilon_i} \tag{6}$$

However, in the context of rainwater infiltration into snow, a more consistent formulation is proposed considering that the frozen rainwater will not become independent ice grains within the snow matrix but alter the snow grains to become indistinguishable with those. As such, the model developed here diverges from previous work in which the forming and melting of ice grains in pores within a soil matrix was considered (cf. Heinze, 2021). For saturated porous media it is well-known that permeability can be linked to porosity using various poro-perm relationships, such as Hazen (1892); Carrier (2003); Kozeny (1927); Carman (1937); Hommel et al. (2018). The well-known Cozeny-Karman relationship relates porosity and hydraulic conductivity

$$K_{sat} = K_0 \frac{\phi^3 \cdot (1-\phi_0)^2}{\phi_0^3 \cdot (1-\phi)^2} \tag{7}$$

with intrinsic hydraulic conductivity $K_0$, porosity $\phi$ (-), and intrinsic porosity $\phi_0$ matching the state of $K_0$. The Cozeny-Karman relationship tends to overestimate the hydraulic conductivity in complex, poorly connected porous media with tortuous flow paths (Mostaghimi et al., 2013) but has been successfully applied for snow in various studies in the past (Albert and Shultz, 2002; Adolph and Albert, 2013, 2014; Meyer et al., 2020). Using Cozeny-Karman, the saturation dependency can be resolved following van Genuchten (1980)

$$K(\psi) = K_{sat}\sqrt{S_{eff}}\left(1 - (1 - S_{eff})^{1/m}\right)^2 \tag{8}$$

The effect of phase change on the hydraulic state of the system is manifold: The amount of water changes as does the hydraulic head due to the source/sink term $M$ in equation 1. The change in hydraulic head will alter the saturation, as will the change in porosity. Subsequently the hydraulic conductivity changes dependent of the changes in porosity and saturation.

## 2.2 Multi-phase heat transfer

Applying the heat transfer model to ROS events, there are three phases to be considered: the snow forming an immobile porous matrix out of ice, liquid water moving relatively to the snow matrix, and air, mainly being replaced by the liquid water. The volume fractions of air and water have to add up to the porosity $\phi$ at all times $\phi = \epsilon_a + \epsilon_l$, and the volume fraction of the ice can then be written as $\epsilon_i = (1 - \phi)$. Each of these phases is described by its own heat equation.

The immobile snow only experiences conduction as heat transport mechanism. However, it can experience internal heat sources or sinks through the phase change $Q_{pc}$ and through heat exchange with water $Q_{il}$ or air $Q_{ia}$. Assuming that density and heat capacity of the snow remain constant, the conservation of energy can be written as:

$$\dot{\epsilon}_i\rho_i C_{p,i}T_i + \epsilon_i\rho_i C_{p,i}\dot{T}_i = \nabla\left(\epsilon_i\lambda_i\nabla T_i\right) + Q_{pc} + Q_{il} + Q_{ia} \tag{9}$$

with thermal conductivity $\lambda$ (W/(m K)). Compared to the heat transfer in frozen soil, the phase change term $Q_{pc}$ in the solid phase is new because in previous models freezing/melting was considered in a separate phase ice to be distinguished from the soil (cf. Heinze, 2021).

The liquid water has advective, conductive and dispersive heat transport components and exchanges heat with snow $Q_{il}$ and air $Q_{la}$. Similarly to snow, it is also affected by phase change $Q_{pc}$. Assuming again that density and heat capacity remain constant, the conservation of energy can be written as:

$$\dot{\epsilon}_l\rho_l C_{p,l}T_l + \epsilon_l\rho_l C_{p,l}\dot{T}_l = -\nabla\left(\epsilon_l v\rho_l C_{p,l}T_l\right) + \nabla\left(\epsilon_l\left(\lambda_l + D_l\right)\nabla T_l\right) - Q_{pc} - Q_{il} + Q_{la} \tag{10}$$

with flow velocity $v$ and thermal dispersion coefficient $D = \alpha_T v$ with dispersive length $\alpha_T$. Applying the chain rule of differentiation and considering the conservation of mass, the equation can be simplified to (Heinze and Blöcher, 2019; Heinze, 2021)

$$\epsilon_l\rho_l C_{p,l}\dot{T}_l = -\epsilon_l v\rho_l C_{p,l}\nabla T_l + \nabla\left(\epsilon_l\left(\lambda_l + D_l\right)\nabla T_l\right) - Q_{pc} - Q_{il} + Q_{la} \tag{11}$$

The air behaves similarly to the water and as the water replaces the air, a similar flow velocity and dispersion coefficient can be assumed. This simplifying assumption of equal flow velocities is consistent with the general model design applied here,

such as assuming incompressibility of water and air, the capillary tube model, and excluding mixture flow within one capillary tube. Hence, if water replaces air during infiltration, the conservation of mass requires the same flow velocity in a tube of constant diameter. In a complex three-dimensional pore structure the flow paths of the air are tortuous and across multiple capillary tubes which cannot be represented here. However, due to the negligible thermal influence of the air, this simplifying assumption has no impact on the simulations' outcome. Further, the air does not experience phase change.

$$\epsilon_a \rho_a C_{p,a} \dot{T}_a = -\epsilon_a v \rho_a C_{p,a} \nabla T_a + \nabla \left( \epsilon_a \left( \lambda_a + D_a \right) \nabla T_a \right) - Q_{ia} - Q_{la} \tag{12}$$

The phases exchange heat based on Newton's law of cooling

$$Q_{ij} = h_{ij} A_{ij} \left( T_j - T_i \right) \tag{13}$$

with the heat transfer coefficient $h$ (W/(m$^2$ K)) and specific surface area $A$ (m$^{-1}$). The subscripts $i, j$ refer to the two phases interacting for each heat exchange term. Hence $i, j \epsilon i, l, a$, and there is a separate heat transfer term for each pair of phases exchanging heat. The heat transfer coefficient $h$ is a parameter, which for porous media is known to depend on flow velocity and grain size (Nield and Bejan, 2013). For various engineering and aquifer materials, a number of (semi-)empirical formulas have been derived based on extensive laboratory and experimental datasets but for snow or ice neither were experiments conducted nor does an analytical formula exists (Wakao et al., 1979; Roshan et al., 2014; Gossler et al., 2020). In this work, the heat transfer coefficient is set constant and varied systematically to assess a possible value range, based on values successfully used for frozen soil derived from the most-general model of Wakao et al. (1979) (Heinze, 2021). It is known that the heat transfer coefficient depends on the flow velocity, which changes dynamically in the described scenario, and hence the heat transfer coefficient will most likely also change. However, there is no experimental data available on the heat transfer coefficient between liquid water and snow, so a constant value is assumed for simplicity. The specific surface area on the other hand is purely based on geometrical considerations. For spherical grains, the specific heat transfer area $A$ is given based on grain diameter $d$ and porosity $\phi$

$$A_{ij} = \frac{6 \left( 1 - \phi \right)}{d}. \tag{14}$$

Snow grains can have various shapes of which "rounded" is one following the international classification for seasonal snow on the ground (Fierz et al., 2009). Rounded snow grains are e.g. caused by repeated melt and freeze processes or blown round from wind at the surface. In general, snow grain shapes and their interactions are very complex and go beyond the scope of this work. For unsaturated conditions, the volume fractions of the phases inside the pores, here liquid water and air, needs to be considered because the contact area between the grain and the separate phases is split between the phases. It has been shown that the contact area available for each individual phase is directly proportional to the saturation of the phases for capillary tube models (Heinze and Blöcher, 2019). Hence, the saturation of liquid water and air is a multiplicator for the respective heat transfer area.

The heat transfer area between water and air within a porous structure can be considered negligible compared to the contact area between water and air with the respective porous matrix (Heinze and Blöcher, 2019). Therefore, heat transfer between infiltrating water and the air phase is neglected here and $Q_{la}(x,t) = 0 \; \forall x, t$.

## 2.3 Considering phase change

In the context of this work, snow is considered as a combination of interconnected spherical ice grains forming a porous matrix characterized by common parameters, such as porosity and permeability. However, the pore structure of snow can differ significantly from those of soils, e.g. indicated by a porosity of $60\%$ or more which is above the limit of a cubic packing possible for equally sized spheres. As pointed out above, freezing and melting processes in the snowpack can result in various crystalline structures of the snow grains (Fierz et al., 2009). For simplicity, in this work we assume that freezing of water will increase the radius of the snow grains, while melting will decrease the radius of the ice grains based on the added or removed volume fraction. We also neglect the specific arrangement of ice grains and possible contact areas. In principle, both processes can occur simultaneously but spatially separated inside a snowpack with complex thermal gradients or a heterogeneous distribution of snow properties, such as snow density and snow morphology.

We use a predictor - corrector scheme to describe the phase changes between liquid and frozen water. In the predictor step, phase change is neglected and the predicted temperature is calculated based on the equations outlined above. If the respective phase temperature, liquid water for freezing and snow for melting, is below or above the respective temperature for phase change $T_{pc}$, the corrector step is applied. In the corrector step, the phase temperature is returned to $T_{pc}$ and the excess temperature is used for the phase change. There are a couple of important notes: (1) The temperature of phase change $T_{pc}$ might experience hysteresis between melting and freezing processes but commonly snow pores are considered too large to experience freezing point depression. The temperature of phase change might also change over time as atmospheric conditions or the pore and grain structure of the snow change. The derived model could incorporate these changes in principle. However, little is known about those dynamics in snow and a deterministic description is complex or even lacking, so that $T_{pc}$ for melting and freezing is kept constant and similar at $0\,°\mathrm{C}$. (2) The predictor - corrector scheme can be numerically cumbersome for small changes in temperature or volume. Therefore, the introduction of a tolerance region around $T_{pc}$ can be numerically necessary (Heinze, 2021). However, this numerical issue does not affect the physical derivation presented here.

In the case of melting of snow grains, the excess thermal energy $\dot{Q}_m$ (W/m$^3$) per discrete time step $dt$ (s) can be calculated as (Kelleners et al., 2016; Heinze, 2021) if $T_i > T_{pc}$

$$\dot{Q}_m = \rho_i C_{p,i} \left( T_i - T_{pc} \right) / dt. \tag{15}$$

The volume fraction of ice that can be melted per unit time with this amount of thermal energy $\epsilon_m$ (-) can be calculated as

$$\dot{\epsilon}_m = -\frac{\dot{Q}_m}{\rho_i L_f}, \tag{16}$$

with the latent heat of fusion $L_f$ (J/kg) (Kelleners et al., 2016). The phase change triggers various subsequent processes. The melted ice only has a temperature of $T_{pc}$. Therefore, it will be warmed to $T_l$ while mixing with the other available liquid water. From this, the water temperature might be decreased by

$$Q_{pc} = \epsilon_m \rho_i C_{p,w} \left( T_l - T_{pc} \right), \tag{17}$$

as $\epsilon_m \rho_i$ is the mass of liquid water considering the density contrast between ice and liquid water. The amount of liquid water can also be altered by the infiltration process in this model and possibly also by other processes such as evaporation, neglected here. As $\epsilon_i = (1 - \phi)$, the change in porosity due to melting is $\dot{\phi} = -\dot{\epsilon}_m$. The change in saturated hydraulic conductivity can then be subsequently calculated using equation 7. Following the change in porosity and liquid water volume fraction, there is also a change in water content and saturation. Further, the additional liquid water needs to be considered as a mass source in the conservation of mass in the hydraulic infiltration model through term $M$ in equation 1. The change in ice grain radius is the factor $\sqrt[3]{\dot{\phi}}$ assuming spherical grains, from which the change in contact area $A$ can be calculated. In principle, it is possible that the infiltration behavior described by the van Genuchten parameters $\alpha$, $n$, $m$ also change during the freezing and melting of the snowpack. The parameters are known to vary for different soil types and soil grain sizes (Schaap et al., 2001). Freezing experiments in soil did not show any indications of an influence of the presence of ice on these parameters (Hansson et al., 2004; Watanabe and Kugisaki, 2017a; Heinze, 2021) but such dynamics have not been experimentally studied for snow so far.

The freezing process is described in a similar way and with the same predictor-corrector scheme. The available energy for freezing once $T_l < T_{pc}$ is calculated as (Kelleners et al., 2016; Heinze, 2021)

$$\dot{Q}_f = \rho_l C_{p,l} \left( T_{pc} - T_l \right) / dt. \tag{18}$$

The volume fraction of water that will freeze per unit time with this amount of thermal energy $\epsilon_f$ (-) can be calculated as

$$\dot{\epsilon}_f = -\frac{\dot{Q}_f}{\rho_i L_f}. \tag{19}$$

Similarly to the possible cooling of the liquid water at melting, the existing ice might get heated by the newly generated ice with temperature $T_{pc}$. This is expressed through the term $Q_{pc}$ in the conservation of energy equation presented above

$$Q_{pc} = \epsilon_f \rho_i C_{p,i} \left( T_i - T_{pc} \right). \tag{20}$$

Changes in porosity, hydraulic conductivity, and grain size radius apply for freezing similar to the melting process described above.

In the proposed model, porosity and grain radius are set independently and the packing is not specified. Ice grains within the snowpack can have a highly irregular sorting, enabling high porosity values of snow above 60%. During the phase change, the mass and volume occupied by the ice grains are altered. To represent the effects of this on the specific surface area of the snow, the ice grain radius is recalculated according to the change in volume fraction of the ice. For an analytically solvable expression, the grains are assumed spherically and the contact area between individual grains is neglected. The updated grain radius after phase change in dependence of the change of the snow volume fraction can be calculated as

$$R_{new} = \sqrt[3]{\frac{1 - \phi_{new}}{1 - \phi_{old}}} R_{old}. \tag{21}$$

Using the density of ice of $917\,\mathrm{kg\,m^{-3}}$ and typical grain sizes of $1\,\mathrm{mm}$ to $3\,\mathrm{mm}$, the range of snow densities between $100\,\mathrm{kg\,m^{-3}}$ and $800\,\mathrm{kg\,m^{-3}}$ results in porosity between $13\,\%$ and $90\,\%$, very reasonable for snow (Kinar and Pomeroy, 2015;

Wang et al., 2017). Melting of snow and freezing of liquid water affect porosity, permeability, and heat transfer area of the snow. In frozen soils, ice grains also might block pores to alter the same parameters. However, the respective relationships are significantly different because the porous soil matrix remains the same and the fractions of the pore filling change, while in snow the porous matrix itself changes (cf. Heinze, 2021). The assumption of spherical snow grains is obviously a strong limitation of the model but enables a consistent mathematical formulation also accounting for growth and decline of snow grain diameter. The spherical shape is used in the model to calculate the surface area of the snow for the heat exchange terms and to estimate the van Genuchten parameters describing the infiltration behavior. The parameters of snow in this work are chosen independent of possible snow crystal structures common for those values for the sake of simplicity. The assumption of spherical snow grains might apply best for previously wetted 'ripe' snow that also experienced grain growth (Colbeck, 1979; Raymond and Tusima, 1979).

## 2.4 Deriving a local thermal equilibrium model

The theoretical framework presented above can be simplified to achieve a comparable LTE model. Mixture theory is applied in the LTE model, not distinguishing between phase temperatures and obtaining thermal parameters based on volumetric weighting as described in Nield and Bejan (2013). Hence, the heat equation solved simplifies to

$$\rho_m C_{p,m} T_m = -\epsilon_{al} \rho_{al} C_{p,al} \nabla T + \nabla \left( \epsilon_m \left( \lambda_m + D_m \right) \nabla T_m \right) - Q_{pc}, \tag{22}$$

with index $m$ denoting the mixture of all three phases, and $al$ the mixture of air and liquid water. The solution procedure regarding phase change remains the same as described above applying a predictor-corrector scheme. In case $T_m > T_{pc}$, additional heat is used to cause melting of the ice and $T_m$ is corrected to $T_{pc}$. Also the water infiltration modeling remains the same than the for the LTNE model

Comparing the newly developed LTNE model to a similarly constructed thermal equilibrium model eliminates other potential influencing factors, such as numerical implementation, choice of parameters, handling of phase change, etc. Instead, it allows to singularly study the effect of averaging phase temperatures in a LTE approach versus individual heat equations in a LTNE model. As described above, there are various substantially advanced LTE models available to describe water infiltration into a snowpack. However, in the light of the research questions addressed in this work, a cross-model comparison is out of scope.

## 2.5 Numerical implementation and tested scenarios

To address the two separate factors on volume fractions, phase change, and infiltration, Dall'Amico et al. (2011) introduced a splitting algorithm separating advective mass flux and phase change. A similar scheme is adopted here, first calculating the liquid water content based on the hydraulic flow given through equation 1. The solution process of the Richards equation is widely described in literature (e.g. Farthing and Ogden, 2017) and usually a two-step procedure, such as the Crank-Nicolson method, is recommended to account for the coupling between saturation and hydraulic conductivity. However, the explicit thermal calculations require a very small temporal resolution $dt$, so that changes in saturation due to infiltration are comparably small and a simple one-step numerical scheme is sufficient (Heinze and Hamidi, 2017; Heinze, 2021). Subsequently, the

**Table 1.** Parameters used in the numerical simulations.

|  |  | ice (i) | liquid water (l) | air (a) |
|---|---|---|---|---|
| $\rho$ | | $917\,\mathrm{kg\,m^{-3}}$ | $1000\,\mathrm{kg\,m^{-3}}$ | $1.2\,\mathrm{kg\,m^{-3}}$ |
| $C_p$ | | $2040\,\mathrm{J\,kg^{-1}\,{}^{\circ}C^{-1}}$ | $4200\,\mathrm{J\,kg^{-1}\,{}^{\circ}C^{-1}}$ | $1008\,\mathrm{J\,kg^{-1}\,{}^{\circ}C^{-1}}$ |
| $\lambda$ | | $2.2\mathrm{W\,m^{-1}\,{}^{\circ}C^{-1}}$ | $0.5\mathrm{W\,m^{-1}\,{}^{\circ}C^{-1}}$ | $0.024\mathrm{W\,m^{-1}\,{}^{\circ}C^{-1}}$ |
| $\eta$ | | - | $1.7\cdot10^{-3}\,\mathrm{Pa\,s}$ | $1.7\cdot10^{-5}\,\mathrm{Pa\,s}$ |

thermal predictor step for each phase is calculated and a violation of the respective physical boundaries $T_l \in [T_{pc},\ T_{boiling}]$ and $T_i \in [0\,^{\circ}\mathrm{C},\ T_{pc}]$ is checked. If necessary, a corrector step for the phase change is conducted as described above. The hydraulic and thermal parameters affected by the phase change are updated and the hydraulic values are calculated again with the updated values to start the calculation of the next discrete time step. Special care has to be taken if the ice content decreases below a critical value so that the snowpack does not act as a porous medium anymore and as such derived governing equations do not apply anymore. Further, mechanical collapse or compaction of the snowpack could occur during a rain event or if melting at deeper snow layers occurs (Bertle, 1966). The mechanical compaction of the snow is partly based on the weight of the snow and the rainwater infiltrating but there are also changes of the crystal and grain structure (Marshall et al., 1999). The cohesive bonds between ice grains account for the snow strength but drops significantly if these bonds are altered (Barraclough et al., 2017). A model of water percolation through a snowpack including compaction of the snowpack has been presented by Meyer and Hewitt (2017). Melting might occur, as seen in the simulation result discussed here, in deeper layers of the snowpack and not necessarily at the surface. Hence, changes in the snowpack structure might cause collapse due to the load above. The simulations are terminated once melting conditions establish within the snowpack and the mechanical failure of the snowpack is to be expected due to an increase of porosity.

The numerical implementation of the governing equations is based on an explicit finite difference scheme. The head-based form of the Richards equation is used to account for possible strong heterogeneity of hydraulic conductivity within the snowpack (Farthing and Ogden, 2017). Water infiltration due to rainfall with intensity $R_i\ \mathrm{m\,s^{-1}}$ into the snowpack at the top boundary condition is formulated as given in Mathias et al. (2015)

$$\psi_1 = \psi_2 - dz + \frac{R_i}{K(\psi_1)} \cdot dz, \tag{23}$$

with subscripts 1 and 2 indicating first and second node point and $dz$ indicating spatial grid distance. The numerical model has been compared to Hydrus 1D solutions and was used in previous works (e.g. Heinze, 2021). The heat equations with advective and diffusive-dispersive fluxes are solved using a third-order upwind scheme and a forward-in-time-centered-in-space finite difference (FTCS) scheme. The numerical results have been tested to not be affected by spatial or temporal resolution within meaningful ranges. The shown results were conducted with a spatial resolution of $0.1\,\mathrm{cm}$. The thermal and hydraulic parameters for water, air, and snow are provided in Table 1. Thermal and hydraulic boundary conditions as well as initial conditions vary with the tested scenarios presented below.

## 2.6 Methods of comparison to historic field data

For testing the ability of the developed model, the field observations from a midwinter rain presented in Conway and Benedict
(1994) are numerically reproduced using the presented LTE and LTNE models. The original data covers a 10-hour-long rain
event with a total amount of 19 mm rainfall on a horizontal snowpack at 915 m elevation in the Cascade Mountains, Washington
(USA), starting at 9pm on the 15th January 1992. The snowpack was systematically instrumented with thermoistors and three
layers of snow separated by two ice layers were identified prior to the event. These layers were from top to bottom: a layer of
15 cm snow consisting of partly rounded grains with a diameter of 0.1 mm; an ice layer of 0.5 cm thickness; a layer of 20 cm
snow consisting of rounded grains with a diameter between 0.1 mm and 0.5 mm; an ice layer of 1 cm thickness; followed by
an another layer consisting of rounded grains with a diameter of between 0.1 mm and 0.5 mm. Air temperature is reported to
be above zero, hence set to 2 °C in the simulation and rainwater temperature was set to the same value. The mean temperature
in the snowpack is reported as -0.9 °C, and set to 0 °C at the surface of the snowpack with a linear thermal gradient to -0.5 °C
at 38 cm below the surface of the snowpack in the simulation. While the original measurements reveled preferential flow
within the snowpack, the presented 1D model focuses on the mean vertical thermal evolution and fluid dynamics. The original
observations show that within the first hour the snow above the first ice crust was warmed to 0 °C. The first and shallowest ice
layer was penetrated 4 hours after the rain onset as the snowpack above was almost fully saturated. Once the upper ice layer
was penetrated, the second and deeper ice layer was reached in less than 15 minutes. The second ice layer was not penetrated
during the observational time. Field observations revealed further that within the first 4 hours only minimal freezing occurred
and no freezing occurred at later times.

   The numerical model represents the upper 38 cm of the snowpack and the first 5 hours after the onset of the rain simulated
as no subsequent hydraulic changes were reported in the original manuscript which could be used to compare the numerical
model and the observation. The initial setting of the snow includes thermal equilibrium between phases within the snowpack
with a small residual volumetric water content of 0.001 across the whole snowpack. Initial porosity and hydraulic conductivity
were set to 0.005 for the ice layers, to 0.05 the upper snowpack layer, and to 0.03 for the lower intermediate snowlayers. The
porosity values were determined from the information that upper snowpack layer was fully saturated after 4 hours of rainfall
assuming a homogeneous rainfall intensity. With a precipitation of 8 mm within the first hour and a height of 15 cm of the
upper snow layer, this results in a porosity of 5 %, which is in the range of the observed water content change (Conway and
Benedict, 1994). Initial hydraulic conductivities were set to match the described temporal evolution of the water front. The
resulting values for the layers from top to bottom were $2 \cdot 10^{-5}$, $1 \cdot 10^{-13}$, $1 \cdot 10^{-4}$, $1 \cdot 10^{-22}$, $1 \cdot 10^{-4}$ m s$^{-1}$, respectively.
The grain radii were set according to the field description. Rainfall infiltration boundary conditions were set to match the field
description.

   The comparison between numerical simulation and field data is done based on significant changes during the ROS event as
described in Conway and Benedict (1994). The provided field data is not suitable for a direct comparison of temperatures, as
depth-resolved observational temperature profiles are not given with sufficient precision and range. However, the (i) timing of

penetration of ice layers, (ii) the evolution of the wetting front, (iii) achieving full saturation, and (iv) observations of melting in time and space provide a great number of other quantitative reference points for comparison.

## 2.7 Methods of model comparison

LTE and LTNE models as outlined above are used to reproduce the same field scenario described in Conway and Benedict (1994). The same parameter set is used for both models, such as hydraulic and thermal parameters, ice grain radii, etc. However, LTE models cannot account for different phase temperatures. Hence, some of the boundary and initial conditions need to be adjusted. In the LTNE model, warmer rainfall can simply be accounted for by a respective boundary condition of the water temperature in combination with the hydraulic infiltration model. In the LTE model, the heat of the infiltrating rain needs to be considered as a thermal flux boundary condition increasing the mixture temperature. Advective heat transport is then described by the hydraulic flow model. Also, for simplicity in the comparison, the snowpack was assumed to have an homogeneous temperature of $0\,°\text{C}$ initially to avoid additional complexity to account for freezing point depression allowing liquid water content at sub-freezing mixture temperatures in the LTE model, which is not necessary for the LTNE model due to separate phase temperatures. Both models result in the same output quantities, mainly volumetric fractions of the phases and temperatures. Hence, model results can be compared side by side as shown below.

## 3 Results from numerical simulations

### 3.1 Comparison to field data and a thermal equilibrium model

With an initial thermal gradient and minimal liquid water present within the snowpack, the effect of the rainwater infiltrating the snowpack is clearly visible hydraulically and thermally (Fig. 1). After 4 hours of rainfall, the upper $15\,\text{cm}$ of the snow until the first ice layer is saturated and significant melting occurred in the upper few centimeters. Once the upper ice layer at $15\,\text{cm}$ depth is penetrated, the water quickly infiltrates through the snow layer below until it reaches the second ice layer. The liquid water content in the upper layer is decreasing at this point as water infiltrates the lower layer quickly once the seal is broken. The upper snow layer is warmed to $0\,°\text{C}$ after 1 hour and melting occurs. Once the rainwater reaches the second snow layer, the computed liquid water temperatures show strong oscillations around the numerical limit for freezing and thawing indicating that minor phase changes of freezing and thawing are likely due to the rapid infiltration of the rainwater cooled to $0\,°\text{C}$ by the passage through the upper snow layer. These small fluctuations of the temperature are purely based on the applied numerical predictor-corrector scheme and the necessity to introduce a tolerance region to calculate phase change. They do not represent the actual thermal fluctuations in the snowpack. Over the simulated 38 cm only minor volume fractions of rainwater are freezing in the simulation result, which is in agreement with the field observation stating that less than 2% of the rainwater froze within the snowpack.

A simple analytical model can be used to estimate the melting rate of a snowpack assuming that the snow is at $0\,°\text{C}$. If the infiltrating rain is considered the only heat source triggering snow melt, the thermal energy rate of the rainwater $\dot{Q}_{rain} =$

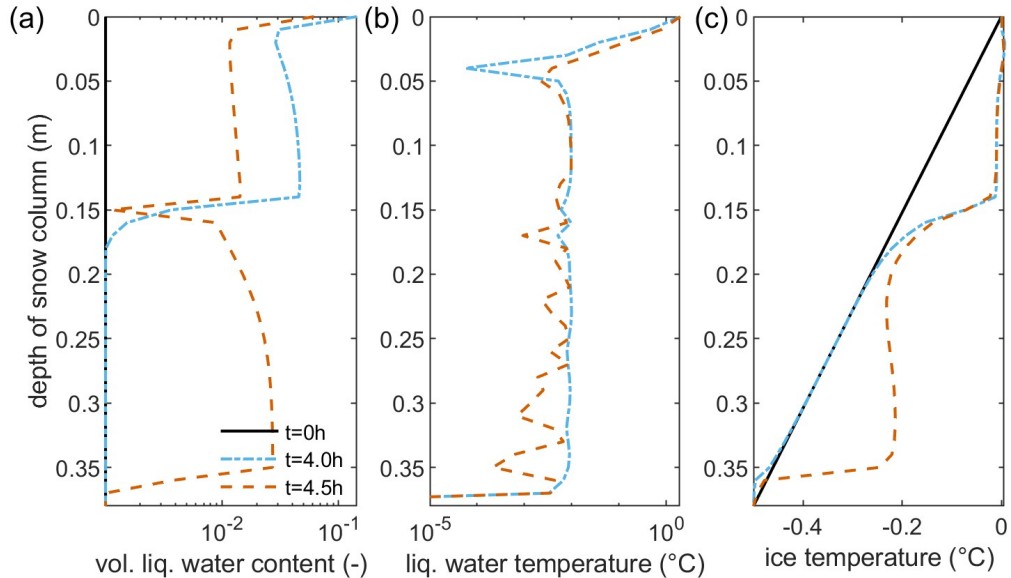

**Figure 1.** Volumetric liquid water content (a), liquid water temperature (b), and ice temperature (c) for the snow column representing the heterogeneous field conditions described in Conway and Benedict (1994) after 0 h, 4 h, and 4.5 h of constant rain. Note the semi-logarithmic scale in (a) and (b) for improved visibility.

$R_i C_p T_{rain}$ is equal to the energy needed for melting $\dot{Q}_{melt} = \dot{m}_{melt} L_f$. As the ratio of $L_f$ and $C_p$ is very close to $80\,°C$, the equation for the melt rate $\dot{m}_{melt} = R_i T_{rain}/80\,°C$ can be obtained. As in the presented case, melting is primarily observed
close to the surface of the snowpack where the snow temperature is close to $0\,°C$, the numerical model results can be compared with this simplified model as an approximation neglecting the presence of the assumed thermal gradient in the snowpack. For the presented field case, this results in approximately $0.38\,\mathrm{kg\,m^{-2}}$ snow melted within the first 4 hours of the rain event. The simulation predicts up to $0.33\,\mathrm{kg\,m^{-2}}$ of snow melted within the first 4 hours of the rain event. The smaller mass predicted by the simulation is dedicated to the applied thermal gradient in the simulation.
The effect of rain intensity on the thermo-hydraulic response of the snowpack during ROS events is of the utmost importance because besides air temperature, the rain intensity is one of the best observed meteorological quantities. In the light of climate change, a tendency towards higher rain intensities also for ROS events is suspected (Juras et al., 2021). To study the effect of rain water intensity on the thermo-hydraulic state of the snowpack, the intensity of the field experiment described above is varied. Rain intensity is set to 0.9, 3.8, and $19\,\mathrm{mm\,h^{-1}}$, hence half, twice, and ten-fold the observed precipitation by Conway
and Benedict (1994). The overall behavior of the snowpack is similar at all four tested precipitation rates (cf. Figs. 1, 2). The lowest tested precipitation rate is not sufficient to cause melting of the upper ice layer and warming of the snowpack to $0\,°C$ is also delayed and reached after 4.5 to 5 hours compared to 1 hour at $1.9\,\mathrm{mm\,h^{-1}}$. The increased precipitation rates provide sufficient water to avoid draining of the upper snow pack once the barrier of the first ice layer is overcome. Naturally, higher precipitation rates also lead to faster saturation and warming of the upper snow layer as well as a faster infiltration into the

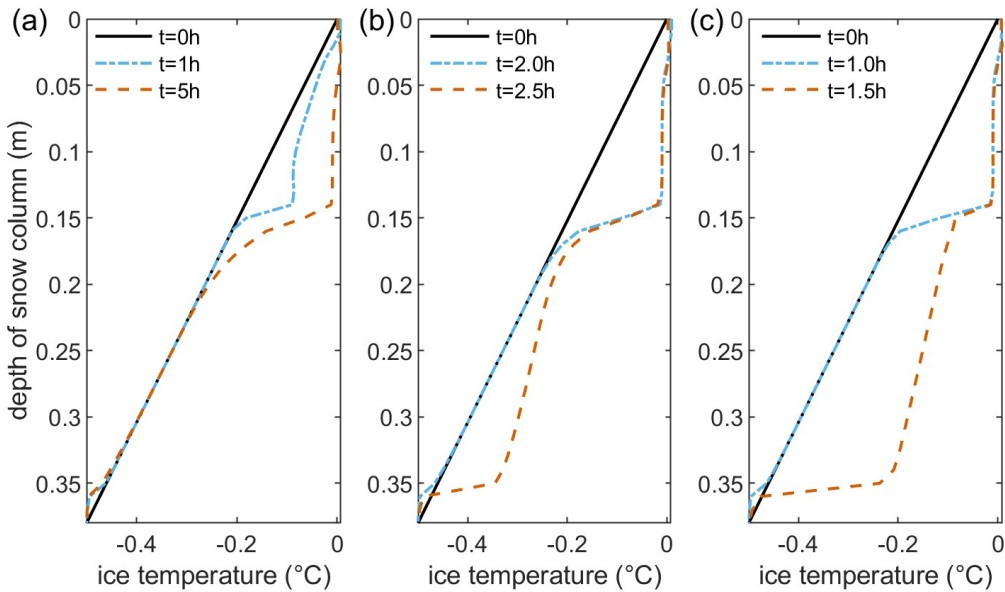

**Figure 2.** Temperature of the ice calculated for the described field scenarios with precipitation intensities of $0.9\,\mathrm{mm\,h^{-1}}$ (a), $3.8\,\mathrm{mm\,h^{-1}}$ (b), and $19\,\mathrm{mm\,h^{-1}}$ (c).

second snow layer. The first ice layer starts melting at around 2 hours for $3.8\,\mathrm{mm\,h^{-1}}$ and after around 1 hour for $19\,\mathrm{mm\,h^{-1}}$ of precipitation intensity.

To further showcase the strength of the newly developed model, also a thermal equilibrium simulation was conducted of the same scenario reported by Conway and Benedict (1994). For simplicity, the snowpack was assumed to have an homogeneous temperature of $0\,°\mathrm{C}$ initially. At the top, the heat of the infiltrating rain was added as a flux boundary condition. All thermal parameters of the involved phases were weighted according to their volume fractions. All other initial and boundary conditions and parameters remained unaltered. Due to its warmer initial state and the thermal equilibrium, melting of the first layer and subsequent infiltration into the second snow layer occurs earlier than in the simulation presented above after around 2.5 hours (cf. Fig. 3 and Fig. 1). As hydraulic parameters were matched between the LTNE simulation and the experiment, a better agreement could be achieved with a different set of hydraulic parameters. Besides the earlier time, passage of the ice layer coincides with melting within the ice layer similar to the previous simulation. The hydraulic behavior of the snowpack is barely altered by the choice of the thermal model as the upper layer starts to drain into the lower snow layer. However, due to the assumption of a warmer snowpack initially, no freezing is observed during the ROS event. The great strength of the newly developed LTNE model compared to equilibrium models is the differentiation between phase temperatures, which allows (i) a consistent formulation of boundary conditions, (ii) study the thermal evolution of the involved phases separately. Hence, the LTNE model provides an improved estimation of the snow temperature if there is a thermal gradient present in the snowpack, which requires a finite amount of time to warm to $0\,°\mathrm{C}$.

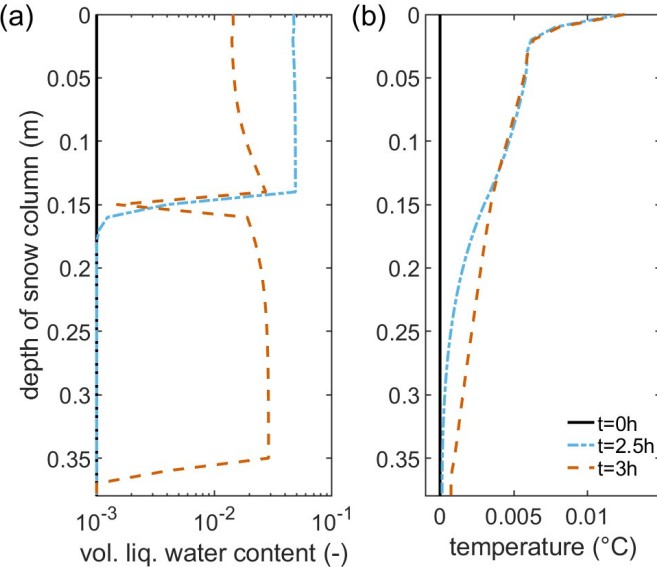

**Figure 3.** Volumetric liquid water content (a), mixture temperature (b) for the snow column representing the heterogeneous field conditions described in Conway and Benedict (1994) after 2 h, 2.5 h, and 3 h of constant rain. Note the semi-logarithmic scale in (a) for improved visibility.

The ice density of the snow grains can vary significantly across snowpacks and within its different layers. To assess the effect of ice density on the simulation results, a parameter variation was conducted for ice densities of $917\,\mathrm{kg\,m^{-3}}$, $940\,\mathrm{kg\,m^{-3}}$, and $1000\,\mathrm{kg\,m^{-3}}$ (Fig. 4). Similar to above, the ice density was assumed homogeneous across the whole modeling domain not separating between the different layers of the snowpack. The higher the ice density, the more thermal energy is needed to cause the same temperature increase. The simulation results show this clearly as for increased ice density, ice temperatures are smaller than for the reference case of $917\,\mathrm{kg\,m^{-3}}$. The effect is not homogeneous across the simulated snow depth but most prominent when heat transfer across the phases occurs, so if the ice temperature is increasing in the simulated scenario. However, the quantitative effect on the calculated ice temperatures are comparably small with less than $0.01\,^\circ\mathrm{C}$.

## 3.2 Generic infiltration into a cold, frozen snowpack

To study the general thermo-hydraulic behavior of water infiltration into a snowpack, a homogeneous snowpack of 50 cm height is assumed with variable thermal gradient and grain size and rain with variable inflow temperature and rain intensity. The temperature gradients in snow can vary significantly depending on the air and ground temperatures at the respective location (Shea et al., 2012). For the study here, moderate thermal gradient with a soil temperature at $0\,^\circ\mathrm{C}$ and an air temperature of -0.1 to $-6.0\,^\circ\mathrm{C}$ are considered (Shea et al., 2012; Wang et al., 2017). Snow porosity is varied between $20\,\%$ and $70\,\%$ (Meyer et al., 2020; Kinar and Pomeroy, 2015), as is the ice grain radius between 1 mm and 3 mm (Clifton et al., 2008). This results in a snow density of $200 - 800\,\mathrm{kg\,m^{-3}}$ (Wang et al., 2017). Hydraulic conductivity of snow covers a wide range (D'Amboise

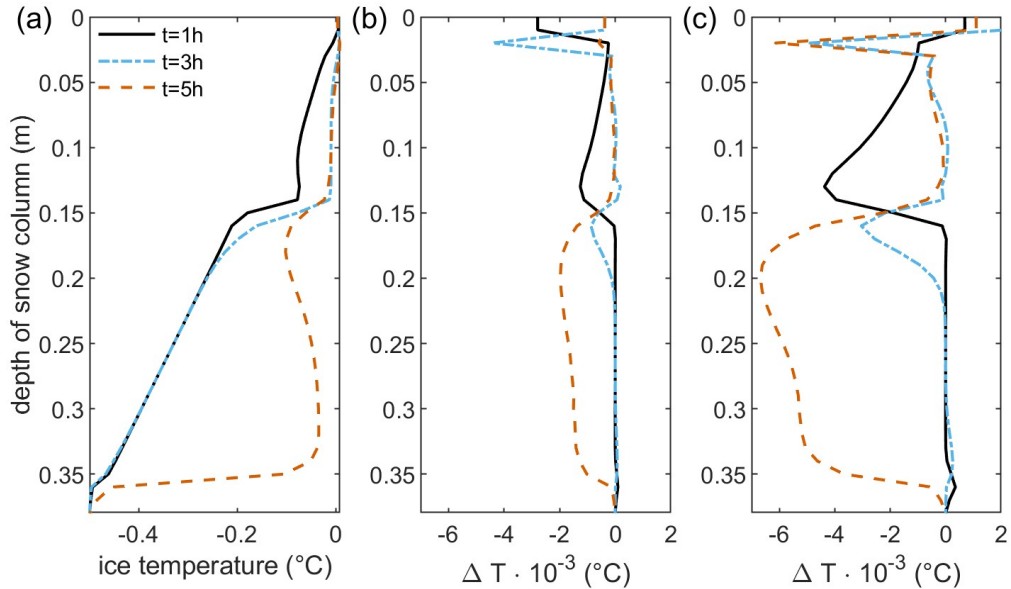

**Figure 4.** Ice temperature for a density of $917\,\mathrm{kg\,m^{-3}}$ (a). Difference in ice temperature for ice density of $940\,\mathrm{kg\,m^{-3}}$ (b) and for ice density of $1000\mathrm{kg\,m^{-3}}$ (c). All simulations were conducted according to the scenario described in Conway and Benedict (1994).

et al., 2017). Here, hydraulic conductivity is varied within a range of $10^{-7}\,\mathrm{m\,s^{-1}}$ to $10^{-3}\,\mathrm{m\,s^{-1}}$. Rainfall temperature is set to moderate or high temperatures in the range of $2\,°\mathrm{C}$ to $4\,°\mathrm{C}$ (Juras et al., 2021). For simplicity the hydraulic boundary condition

at the top is set to a constant $0.1\,\mathrm{m}$ water column analog to comparable soil experiments (e.g. Hansson et al., 2004; Watanabe and Kugisaki, 2017b).

    To study the effect of water infiltration into a cold, frozen snowpack, the first simulations consider an air temperature of -3 °C and a rainwater temperature of 4 °C. The snowpack, therefore, experiences a thermal gradient from initially -3 °C at the surface to 0 °C at the bottom of the snow/soil interface. The influence of several parameters on the thermo-hydraulic processes

within the snowpack is studied based on a systematic parameter variation. In total, five different configurations are studied, with varying values in hydraulic conductivity $K$, porosity $\phi$, ice grain radius $R$, and heat transfer coefficient $h$. Hydraulic conductivity $K$ and porosity $\phi$ are controlling factors for the flow behavior during infiltration, while ice grain radius $R$ affects infiltration due to the dependency of the van Genuchten parameters $\alpha, n$ on $R$ (Eqs. 4 & 5) as well as the heat transfer area (14), and $h$ controls the heat transfer between phases. An overview of the chosen scenarios is given in Table 2.

Scenario A is selected as the baseline for comparison with the other scenarios. In this scenario, melting of the snow in the top $10\,\mathrm{cm}$ can be observed after 12 hours of rainwater infiltration without significant advancement in the following 12 hours. Up to 15 % of the snowpack were melted (Fig. 5a). Changes in the liquid water temperature over time show that a steady state was not reached within 24 hours. The liquid water temperature decreases over time in the upper $20\,\mathrm{cm}$ of the snow column. Initially, the warmer rainwater infiltrates into the snowpack without melting significant amounts of snow. Infiltration is comparably quick

and heat exchange between phases is small, hence, liquid water temperature remains above the temperature of phase transition

**Table 2.** Parameters varied in the five scenarios compared for rainwater infiltration into the snowpack. Scenario A is the base scenario and in the other scenarios one parameter is varied compared to scenario A.

| ID | $R$ [m] | $\phi$ [-] | $K$ [$\mathrm{m\,s^{-1}}$] | $h$ [$\mathrm{W(m^{-2}K^{-1})}$] |
|----|---------|-----------|---------------------------|----------------------------------|
| A | 0.001 | 0.4 | $10^{-4}$ | 0.1 |
| B | 0.001 | 0.4 | $10^{-6}$ | 0.1 |
| C | 0.001 | 0.2 | $10^{-4}$ | 0.1 |
| D | 0.003 | 0.4 | $10^{-4}$ | 0.1 |
| E | 0.001 | 0.4 | $10^{-4}$ | 1 |

for the upper $20\,\mathrm{cm}$. As the snow starts to melt at the top of the snowpack, the liquid water temperature is decreasing during infiltration due to the mixture with meltwater (Fig. 5b). The thermal energy of the infiltrating water is sufficient to warm the snow temperature close to the temperature of phase transition (Fig. 5c). There are several changing points in all variables, which require further discussion. Very close to the top at around $3\,\mathrm{cm}$ there is a notch in the volumetric ice content. This notch

is generated in the first thirty seconds of rainwater infiltration as the infiltrating water is almost immediately melting the top of the snow cover. However, the meltwater cools the infiltrating water close to the temperature of the phase transition that a part of the liquid water is freezing during infiltration a few centimeters later. This frozen water melts again quickly after but the short period of freezing is sufficient to sustain the small alteration in the otherwise smooth trends of volumetric ice content and liquid water temperature. It can be observed that this notch becomes more significant for increased heat transfer mechanisms

in scenarios D and E, as shown below.

Another changing point in the profiles is around $20\,\mathrm{cm}$ along the snow column, as the volumetric ice content is increasing at this depth rapidly reaching $0.5\,\%$ higher ice content than the initial value (Fig. 6a). The ice content then declines toward its initial value at the bottom of the snowpack. This increase in ice content is caused by infiltrating water cooled to the freezing point by meltwater and heat transfer to the snowpack. The frozen water content does not change remarkably between 12 hours

and 24 hours of continuous infiltration but it can be observed, that melting continues above this layer and will reduce ice content for longer times of infiltration. This can also be seen in the liquid water temperature, as the liquid water becomes warmer with time at this depth (Fig. 6b). The ice temperature also increases by $0.05\,^{\circ}\mathrm{C}$ over 12 hours (Fig. 6c). It can therefore be expected that the whole snowpack would melt for ongoing precipitation. The bumpy liquid water temperature profile with a few sharp edges and the corresponding changes in the volumetric ice content can be explained by the numerical algorithm requiring at

least a temperature difference of $0.01\,^{\circ}\mathrm{C}$ to the temperature of phase transition before a phase change can be calculated.

The described processes also affect the snow parameters. As such, the ice grain radius barely increases due to the freezing but is reduced to $0.0008\,\mathrm{m}$ at the top surface. Subsequently, the heat transfer area was reduced to $65\,\%$ of its original value at the top of the snow. The hydraulic conductivity also increased at the top towards $0.0017\,\mathrm{m\,s^{-1}}$, while it declined slightly to $0.00095\,\mathrm{m\,s^{-1}}$ around the height of $23\,\mathrm{cm}$ with the highest volumetric ice content. The hydraulic conductivity then increases

again with depth towards its initial value at the bottom.

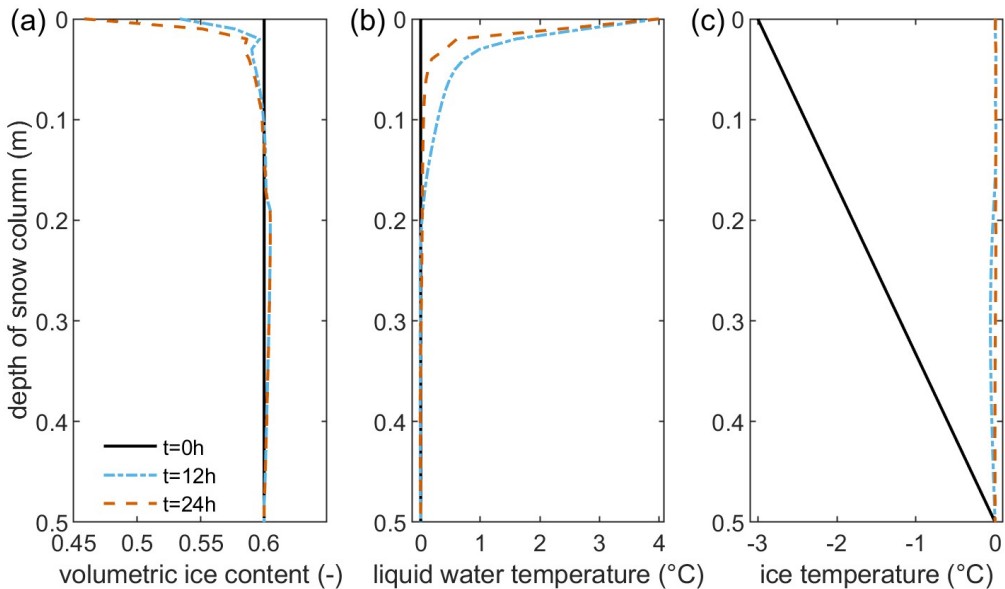

**Figure 5.** Volumetric ice content (a), liquid water temperature (b), and ice temperature (c) for the whole snow column of scenario A for the initial conditions, after 12 hours, and after 24 hours of continuous rainwater infiltration with $4\,°C$ rainwater assuming a constant hydraulic head of $0.1\,m$ at the top boundary.

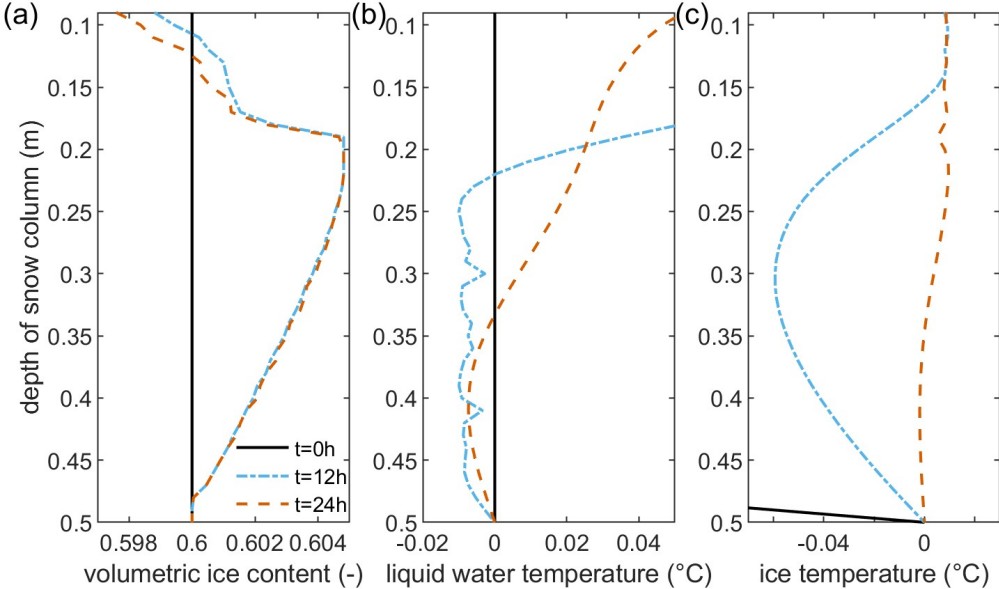

**Figure 6.** Volumetric ice content (a), liquid water temperature (b), and ice temperature (c) for the bottom $40\,cm$ of the snow column of scenario A for the initial conditions, after 12 hours, and after 24 hours of continuous rainwater infiltration with $4\,°C$ rainwater assuming a constant hydraulic head of $0.1\,m$ at the top boundary. Variables are cut within a constrained value range to highlight small variations.

## 3.3 Influence of hydraulic parameters

Hydraulic conductivity influences the temperature evolution in the snowpack during rainwater infiltration most specifically due to the advective part of heat transport. The slower heat advection from the top of the snow cover towards deeper layers affects predominantly the liquid water temperature profile (scenario B - Fig. 7b). In general, due to the fixed pressure boundary condition at the top and bottom of the snow column and due to the reduced hydraulic conductivity, less rainwater is flowing through the snow column in scenario B than in scenario A. Less mass of warm rainwater also means less thermal energy is added to the system. Compared to scenario A, the melting of ice at the top is in a similar range of 15 % of the volumetric ice content after 24 hours. However, the maximum depth where melting occurred is lower after 12 hours and 24 hours respectively in scenario B than in scenario A, and more ice is melted towards the top than towards deeper parts of the snowpack. The liquid water temperature is above the temperature of phase transition until 15 cm after 12 hours, and around 21 cm after 24 hours. This can be explained by the lower hydraulic conductivity increasing the residence time of the infiltrating water close to the surface of the snowpack. Subsequently, deeper parts of the snowpack warm less in scenario B than in scenario A and the ice temperature after 12 hours of infiltration is smaller (Fig. 7c). Remarkably, the temperature difference between phases is more sustainable in scenario B than scenario A (cf. Figs. 7b & c) Due to the melting at the top, the hydraulic conductivity has increased there to $4.2 \cdot 10^{-6} \, \mathrm{m\,s^{-1}}$, while its lowest value is just slightly decreased from its initial value to $9.6 \cdot 10^{-7} \, \mathrm{m\,s^{-1}}$. The snow grain radius decreased to $8.2 \cdot 10^{-4} \, \mathrm{m}$ at the top accordingly but barely increased from its initial value anywhere in the snowpack with a maximum value of $1.001 \cdot 10^{-4} \, \mathrm{m}$ at 21 cm from the top of the snowpack after 24 hours. The grain radius decreases back to its initial value towards greater depth in agreement with a decrease in volumetric ice content for the greater depth of the snowpack. Consequently, the heat transfer area is reduced to 2/3 of its initial value at the surface but remains almost constant anywhere else besides in the top 10 cm. Also remarkably, the spikes in volumetric ice content in the top few centimeters caused by the refreezing of infiltrating and melting water in scenario A do not occur in scenario B. Due to the slower transport of heat and the longer residence time, meltwater is not transported quickly so it gets warmed by conduction from infiltrating water before it can refreeze.

The overall thermo-hydraulic response of the snowpack does not change with a decreased porosity (scenario C - Fig. 8) but several tendencies can be observed. The melting is still focused around a few centimeters at the top but freezing occurs already at around 8 cm from the top and the increase in volumetric ice content is with 0.9 % also larger than in scenario A. Contrary, the amount of melted water at the top is larger with 20 % of snow volume being melted within the first 24 hours. These changes also reflect on the hydraulic conductivity and the ice grain radius. The hydraulic conductivity at the top is significantly increased to $0.0085 \, \mathrm{m\,s^{-1}}$ at the top, while its slightly reduced to $8.5 \cdot 10^{-5} \, \mathrm{m\,s^{-1}}$ in the part of the snowpack where freezing occurred from its initial value of $10^{-4} \, \mathrm{m\,s^{-1}}$. The ice grain radius is barely increasing, as the growth in ice grain radius becomes smaller for higher volumetric ice contents due to the cubic dependence assuming spherical grains. The melting more severely changes the ice grain radius to approximately $8 \cdot 10^{-4} \, \mathrm{m}$ due to the substantial change in volumetric ice content. Subsequently, to the grain radius and the volumetric ice content, the heat transfer area only changes significantly at the top, reducing to roughly 65 % of its initial value. The differences to scenario A can be explained by the lower initial porosity because with lower porosity the

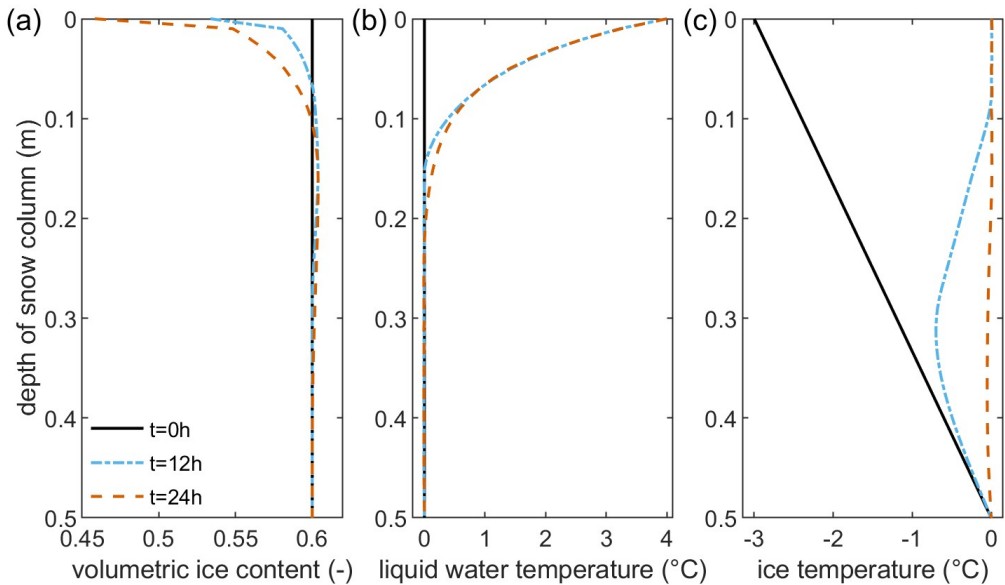

**Figure 7.** Volumetric ice content (a), liquid water temperature (b), and ice temperature (c) for the snow column in scenario B for the initial conditions, after 12 hours, and after 24 hours of continuous rainwater infiltration with $4\,^\circ\text{C}$ rainwater assuming a constant hydraulic head of $0.1\,\text{m}$ at the top boundary.

heat exchange area between water and snowpack is higher in scenario C than in scenario A. Therefore, the infiltrating water transfers more heat to the snow in the top centimeters of the snowpack causing more ice to melt. The melt water cools the infiltrating water in addition to the heat transferred to the snow so that the infiltrating water cools to the temperature of phase transition within a shorter distance from the top in scenario C than in scenario A. This causes the freezing of infiltrating water closer to the surface. However, the temperature evolution within the studied 24 hours of infiltration shows similar trends to scenario A, especially with an increase in snow temperature to the temperature of phase transition across the whole snowpack within 24 hours. This indicates that for prolonged warm rainwater infiltration, the melting of the snowpack would continue from top to bottom but at a slower rate compared to scenario A.

## 3.4 Influence of heat transfer parameters

The relevance of the heat transfer parameters becomes obvious in scenarios D and E in which the ice grain radius $R$ and the heat transfer coefficient $h$ are altered respectively. In scenario D, the variation of the ice grain radius $R$ also changes the infiltration behavior based on equations 4 and 5 as outlined above. The most obvious differences to the profiles of the previously shown scenarios A-C are the strong spikes close to the surface in scenarios D and E (Figs. 9 & 10). These spikes, which in a similar kind can also be observed in the other scenarios, can be explained, as outlined above in the description of scenario A, by processes in the very first seconds and minutes of the infiltration. Due to the reduced heat transfer in scenario D, compared to the previously shown scenarios, these spikes are just significantly more persistent over time. The melt water cools

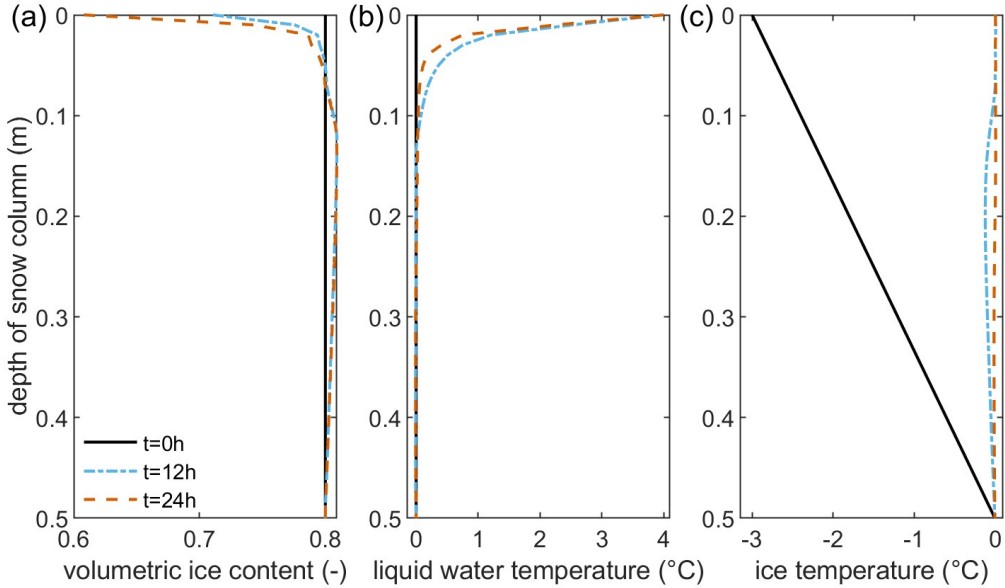

**Figure 8.** Volumetric ice content (a), liquid water temperature (b), and ice temperature (c) for the whole snow column of scenario C for the initial conditions, after 12 hours, and after 24 hours of continuous rainwater infiltration with $4\,°\text{C}$ rainwater assuming a constant hydraulic head of $0.1\,\text{m}$ at the top boundary.

the infiltrating water so that parts of the infiltrating water are freezing again shortly after. Therefore, volumetric ice content is decreasing close to the top but not directly at the top due to additional cooling by air represented in the respective boundary conditions. This causes the first spike to lower volumetric ice content values (Fig. 9a). The freezing of the melted water mixed with infiltrating water shortly after causes the increase in volumetric ice content shown in the second spike. As the warm water

infiltration continues over time, the water frozen at the beginning of the infiltration becomes melted afterward. However, the amount of frozen water causing an increase of $1\,\%$ in volumetric ice content is substantial enough, so that the spikes remain visible, while shifting in their values, even after 24 hours of infiltration. The progressing melting of this additional ice also affects the temperature profile, even after 24 hours (Fig. 9b). The value range showing a maximum decrease in volumetric ice content of slightly more than $4\,\%$ is significantly less in scenario D compared to the other scenarios, further emphasizing

the effect of this initial freezing process. Besides these spikes, the overall thermo-hydraulic processes in the snowpack are comparable to the previous scenarios. However, in scenario D the freezing of the infiltrating water occurs deeper within the snowpack than in scenario A. This is partly a consequence of the freezing and melting processes close to the snowpack top just described but also partly caused by the infiltration parameters $\alpha$ and $n$ in dependence of the initially larger ice grain radius compared to the other scenarios. Across the profile, the changes in volumetric ice content reflect on the ice grain radius varying

its values between $2.826 \cdot 10^{-3}\,\text{m}$ at the top and $3.005 \cdot 10^{-3}\,\text{m}$ at the point of the largest amount of volumetric ice content. The significant changes in volumetric ice content also reflect on the hydraulic conductivity varying in the range of $9.7 \cdot 10^{-5}\,\text{m s}^{-1}$ to $2.8 \cdot 10^{-4}\,\text{m s}^{-1}$. The range in heat transfer area varies between $5.3 \cdot 10^{2}\,\text{m}^{-1}$ and $6.0 \cdot 10^{2}\,\text{m}^{-1}$, roughly a third compared to

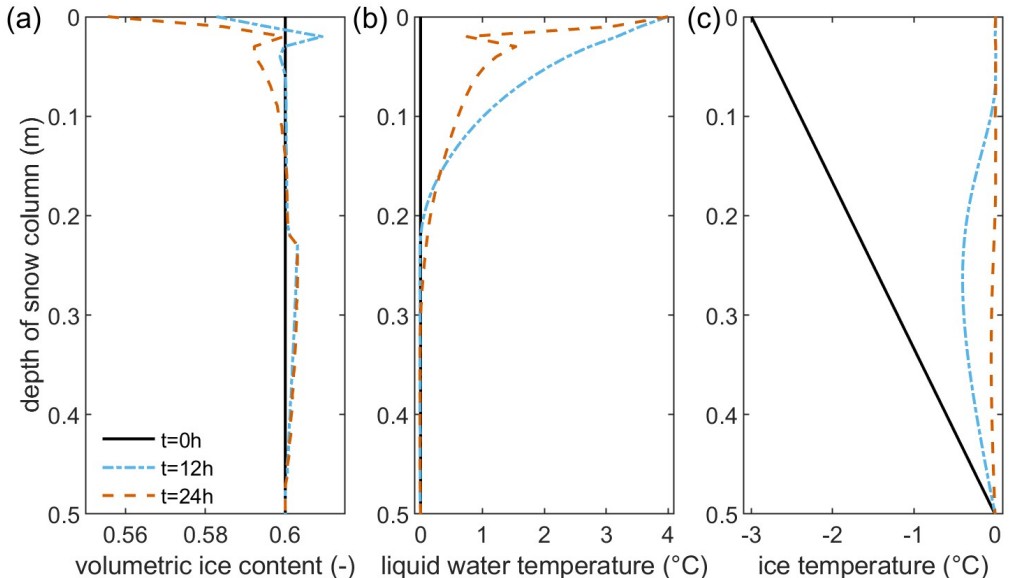

**Figure 9.** Volumetric ice content (a), liquid water temperature (b), and ice temperature (c) for the whole snow column of scenario D for the initial conditions, after 12 hours, and after 24 hours of continuous rainwater infiltration with $4\,°C$ rainwater assuming a constant hydraulic head of $0.1\,m$ at the top boundary.

the values from the other scenarios. The reduced heat transfer between phases is especially visible in the ice temperature after 12 hours (Fig. 9c) with temperatures below $0\,°C$, while in scenario A the ice temperature is already above $0\,°C$ after 12 hours

(Fig. 6c).

Scenario E was conducted with a heat transfer coefficient ten times larger than in the other scenarios and therefore ten times larger as predicted by the only applicable semi-empirical formula by Wakao et al. (1979). It needs to be reminded that this formula, while applicable from a parameter point of view, was neither developed nor tested for snow. The heat transfer coefficient between water, air, and ice is unknown due to a lack of experimental investigation, and the used parameter value is

only the best available guess. Scenario E investigates the influence of the heat transfer coefficient on the melting behavior of the snowpack - and the impact is significant. While, similarly to scenario D, the spikes caused by early freezing processes are sustainable along long time scales, the volumetric ice content melted in scenario E within 24 hours is more than three times the amount than in scenarios A to C. The melting only occurs in the top $6\,cm$ of the snowpack and phase temperatures equalize below at the temperature of phase transition without any phase changes taking place. This shows, that the thermal energy added

to the system through the rainwater infiltration is transferred to the snow within the top few centimeters. Further down in the snowpack, due to the large heat transfer coefficient, both phases reach equilibrium quickly, so that there is no refreezing of melt or infiltrated water at deeper layers. The equilibrium between phase temperatures is to be expected for large heat transfer coefficients. The thermal non-equilibrium at the top remains solely due to the physical limits of phase temperatures requiring thermal energy for the phase transition.

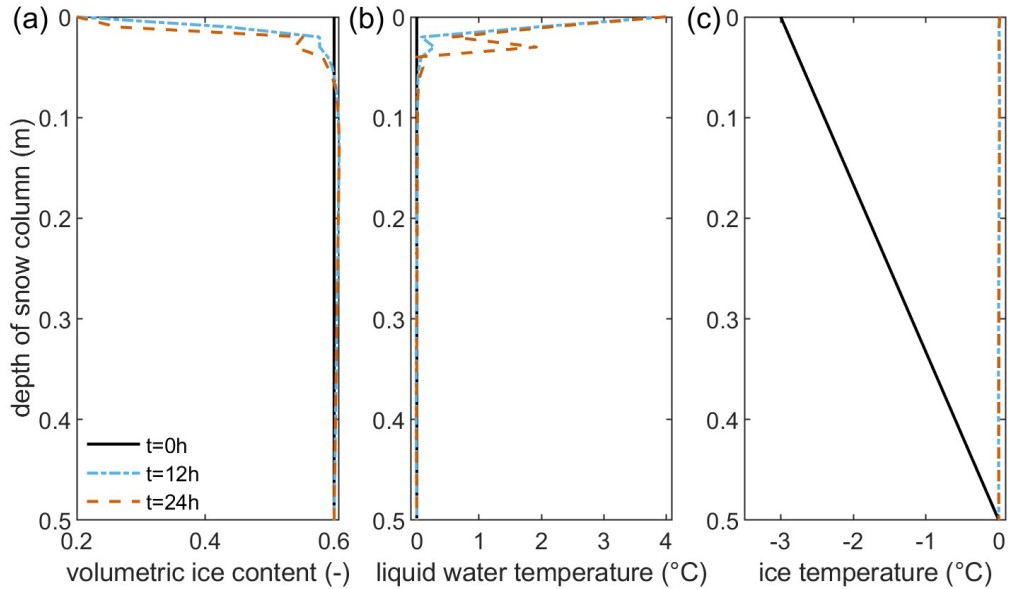

**Figure 10.** Volumetric ice content (a), liquid water temperature (b), and ice temperature (c) for the whole snow column of scenario E for the initial conditions, after 12 hours, and after 24 hours of continuous rainwater infiltration with $4\,°C$ rainwater assuming a constant hydraulic head of $0.1\,m$ at the top boundary.

## 3.5  Accelerated melting of a thawing snow

To study the effect of rainfall on an already thawing snowpack with a small temperature gradient, the simulation considered in this subsection applied an air temperature of just $-0.1\,°C$ and a rainwater temperature of $4\,°C$. The snowpack therefore experienced almost no thermal gradient with depth as the snow/soil interface is assumed to have a temperature of $0\,°C$. The parameter setting is chosen similar to scenario A presented above (Table 2). The overall behavior and the ongoing thermo-hydraulic processes within the snowpack are very comparable to the previously seen results in scenario A (Figs. 5 & 11). Melting occurs solely on the top in the first $20\,cm$ of the snowpack and the previously discussed spikes in the top few centimeters are also observable. In comparison to scenario A, the difference in depth affected by melting is almost similar after 12 and 24 hours, while the maximum change in volumetric ice content is with $15\,\%$ almost similar. Heat is added to the system by the infiltration of warm rainwater, which is subsequently transferred to the snowpack. This heat is almost directly used for melting due to the snow temperature being close to the temperature of the phase transition. Therefore, melting occurs deeper within the snowpack at earlier times than in scenario A, as the warming of the snowpack is omitted. The depth of $27\,cm$ is the limit until all heat is transferred to the snowpack, as can be seen in the ice temperature (Fig. 11c). From the top to $27\,cm$ range, the ice temperature oscillates within the numerical limit around the temperature of phase transition. Below it is balanced by the cooling from the frozen soil within the 24 hours of simulation time. The amount of ice melted is marginal (yet) below $20\,cm$ (Fig. 11a). Melting occurs predominantly in the top centimeters and the meltwater can cause a cooling of the liquid water temperature over time

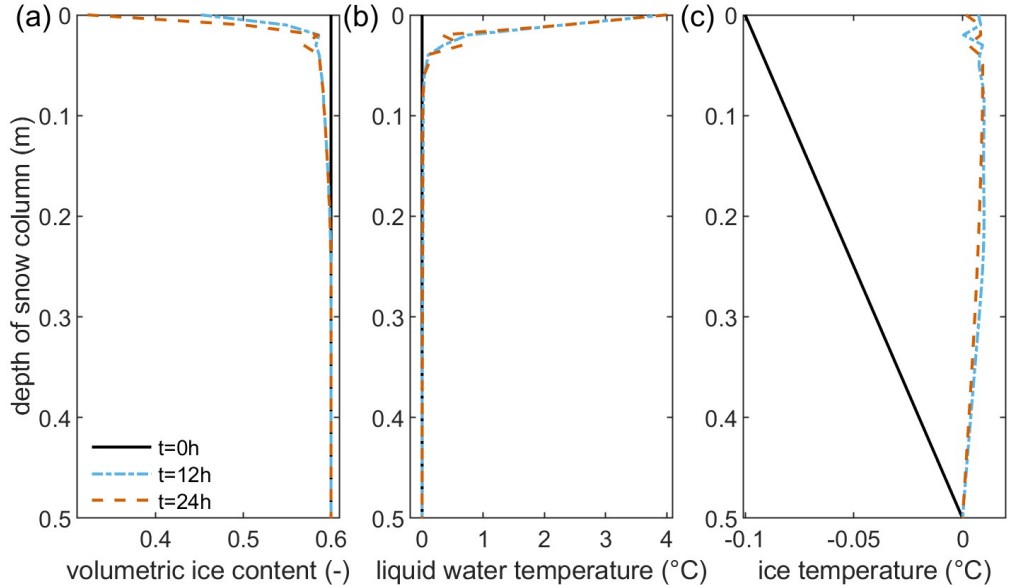

**Figure 11.** Volumetric ice content (a), liquid water temperature (b), and ice temperature (c) for the whole snow column close to thawing for the initial conditions, after 12 hours, and after 24 hours of continuous rainwater infiltration with 4 °C rainwater assuming a constant hydraulic head of 0.1 m at the top boundary.

in the top few centimeters. Due to the snow temperature being close to the temperature of phase transition, the snowpack has almost no cooling capacity, so there is no refreezing of melted water within the deeper layers of the snowpack. This example shows in agreement with previous studies on ROS (e.g. Mazurkiewicz et al., 2008), that the thermal energy of the rain is insufficient to trigger large amounts of snow melting - especially considering the long-lasting rain event simulated here. The warming of the snowpack towards the temperature of phase transition is significantly less energy-consuming than the melting. Therefore, the thermal gradient of the snowpack is less relevant for the overall response of the snowpack to a ROS event than thermal or hydraulic parameters. However, even in this scenario with almost all temperatures close to 0 °C local thermal non-equilibrium between phases persists as the liquid water temperature is above 0 °C in the top 29 cm and the temperature difference to the ice temperature is significantly above the numerical accuracy.

## 4   Discussion

### 4.1   Benefits of applying a LTNE model to simulate ROS events

First of all, the results show the applicability of the LTNE approach for ROS events and that LTNE is persistent over long time scales in all tested scenarios at least partly within the snowpack. In a broader context, the presented work is a first-of-its-kind application of a multi-phase LTNE approach in which the volume fractions of all phases can vary over time. This is a major methodological advancement from previous work, which considered the solid phase stationary during water infiltration into

frozen soil (Heinze, 2021). The warm water infiltrates down to a depth of $30\,\text{cm}$, in most tested cases at least $10\,\text{cm}$, into the snowpack until it reaches the temperature of phase transition and often thermal equilibrium with the snowpack. At shorter time scales, the LTNE between phases can even reach greater depth in the snowpack in the presence of a strong thermal gradient within the snowpack. This is very comparable to the findings for water infiltration into soil presented in Heinze and Blöcher (2019). Besides the coupling between thermal and hydraulic processes, the overall thermal-hydraulic behavior between LTE and LTNE models is comparable. However, the time scales can vary. In the tested scenario the LTE model, with the absence of a thermal gradient in the snowpack, warming of the snowpack and melting of the intermediate ice layer occurs substantially earlier than in the LTNE model. Compared to the LTE model assuming thermal equilibrium between phases, the presented novel model can distinguish phase temperatures. In the tested scenario of the field oberservation by Conway and Benedict (1994) the individual snow layers show temperature differences of $0.2\,^\circ\text{C}$ or more between the snow and the infiltrating water, even hours after the start of the ROS event if intermediate ice layers temporarily block water percolation. In any scenario, the LTNE allows the formulation of consistent boundary conditions considering the physical limits in phase temperatures due to the nature of each phase. Hence, the LTNE model seems the more suitable approach to model ROS events than the thermal equilibrium approach. However, there is some uncertainty regarding the heat transfer coefficient in snow and with substantially larger heat transfer coefficients LTNE models will converge towards a thermal equilibrium between phases more quickly. There is substantial need for further investigations, theoretically and experimentally, to further constrain the parameterization of the modeling approach.

## 4.2 Comparison to field observations and implications for natural hazards

Within the context of ROS events, the model only touches on a few of the many influencing factors. Still, major observations from these events have been reproduced in the simulations above, such as a (re-)freezing or melting close to the surface. Both was observed by Conway and Benedict (1994) at one event during which within the two hours roughly $17\,\%$ of rain froze in the snowpack but vanished afterwards, and melting for the event described above in more detail. The observations also showed, that the heat transport within the snowpack is primarily advection dominated, at least along pathways, similarly to the simulation results. The shown simulations suggest similar to previous rough energy balance calculations (Mazurkiewicz et al., 2008), melting of the snowpack in total through ROS is almost negligible and becomes only relevant under already melting, or at least warming, conditions. The risk perception of slush flows due to ROS events therefore might be overestimated as those occur rarely and usually with clear indicators such as warm air temperatures and significant rainfall over several hours (Mazurkiewicz et al., 2008). Under these conditions, the rainfall itself might already pose a substantial hazard even without the presence of snow. The strength and the additional insights of the presented model come from the depth-resolved description providing detailed information about the freezing and melting of rainwater and snow along the snow profile, as well as the heterogeneous layering of the model providing thermal and hydraulic barriers.

## 4.3 Approaches to further constrain the model parameters

The simulation results emphasize the necessity to experimentally obtain separate phase temperatures to quantify the relevance of LTNE in the field application. The heat transfer coefficient has a significant impact on the thermo-hydraulic evolution in the snowpack and up to date, to our best knowledge, there is not a single experiment studying heat transfer in ice or snow. Existing equations for the heat transfer coefficient have been developed with different applications in mind. On the other hand, ROS experiments might be very suitable to study LTNE effects in porous media, due to the existence of comparably large pores and a flexible porous media suitable for the installation of small temperature sensors. This might facilitate the measurement of water or air temperature separately from the solid phase temperature compared to experiments done in granular soils (Gossler et al., 2019). With the phase separation and depth-resolved information within the snowpack, experimental techniques providing such information become of special interest. This includes the monitoring the movement of the wetting front and fingering using MRI, as well as the changes of the snow microstructure using $\mu$CT (Katsushima et al., 2020). While such techniques are only suitable for the laboratory scale, passive microwave monitoring allow the observation of freezing and melting in the snowpack at a field site (Cagnati et al., 2004). Such non-destructive monitoring techniques do not provide depth-resolved information but do allow estimation of the liquid water stored within the snowpack. Resolved sufficiently fine, such methods can be used to further constrain and validate the presented model.

## 4.4 Future developments towards more realistic snowpack representation

A snowpack is typically layered with possibly strong contrasts in thermo-hydraulic parameters between snow layers due to the different snow genesis. Of special importance are preferential flow paths within the snowpack, vertically as well as horizontally along stratification layers, which are often seen in dye tracer experiments (Stähli et al., 2004; Juras et al., 2017). However, as those preferential flow paths within the snowpack have similarities to (micro-)fractures within the porous media, the flow behavior as well as the heat transfer along preferential flow paths might be different compared to the porous snow matrix. Relevant modeling approaches, based on local thermal equilibrium, include a full continuum-mechanical three-dimensional approach (Hirashima et al., 2017), a dual-domain approach (Würzer et al., 2017), or lagrangian mechanics (Ohara, 2024). It remains a future field of study, if and to what extent known concepts of describing heat transfer in a multi-phase environment in fractured porous media, can be transferred from rock and soil toward snow and ice.

In the presented approach the infiltration behavior, in terms of the van Genuchten parameters $\alpha$, $n$, was not altered during freezing and melting, but this might be the case in a realistic snowpack as the parameters depend on the grain size of the ice particles. However, neither was such effect experimentally studied to see if the effect is of relevance nor can this effect be easily included in a numerical model as such dynamics require special numerical handling due to their effect on the hydraulic pore pressure calculations and the conservation of mass. The model also only considered rounded ice grains and any other shape and condition of snow grains, any snow stratification or snow grain metamorphosis was neglected. Especially changes in snow density, crystal structure, the forming of preferential flow paths and fingering have not been considered (e.g. Marshall et al., 1999). These can be crucial to reproduce the realistic response of a snowpack to rain-on-snow events and its potential

impact to natural hazards. To achieve this in a realistic setting, several model extensions are required. As such, different snow morphology in the layers needs to be considered, e.g. through coupling with the software SNOWPACK. Further, atmospheric influences, such as snow albedo and wind speed, need to be considered as they influence the thermo-hydraulic state at the top of the snowpack. Additional parameters about the rainfall, such as drop size and speed, might also affect the system response at the top of the snowpack with consequences for the whole underlying snow. Similarly, multiple variables of the soil have not been considered in this work, of which many might have a strong influence on the freezing and melting behavior of the rainwater within the snowpack, as well as controlling its discharge at the snow/soil interface. In the future, also geometrically more advanced models will be necessary to account for a possible surface runoff on top of the snowpack on an inclined hill slope, horizontal flow, as well as variation in snowpack thickness and compaction due to surface morphology, vegetation, etc.

## 5  Conclusions

This work presents the derivation of a true multi-phase LTNE approach in which the volume fractions of all phases can vary over time and its application to Rain-on-Snow events. The simulation results indicate that depending on the initial thermal and hydraulic structure of the snowpack, differences of the phase temperatures can persist over hours and over decimeters of snow depth. This suggests that LTNE models might be the more suitable choice than equilibrium models to adequately describe the thermal evolution within a snowpack during ROS events if the snowpack exhibiting a thermal gradient initially. However, uncertainty remains regarding model parameterization especially regarding the heat transfer coefficient and for realistic applications preferential flow paths within the snowpack need to be considered.

*Code availability.*  The source code is available at https://gitlab.com/thomhGeoCode/ltnesnow

*Author contributions.*  This is a single author manuscript

*Competing interests.*  The author declares that there are no competing interests.

*Acknowledgements.*  The author appreciates funding through the German Research Foundation (DFG) with grant HE 8194/4-1.

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
