# Peer review of "A local thermal non-equilibrium model for Rain-on-Snow events"

_Hydrology and Earth System Sciences, 2024_

## Author Comment (AC1)

Response to Reviewer Comments #1 (**Responses in bold**)

**Thank you for taking the time to conduct this thorough review and for your constructive comments, which will be used to substantially improve the manuscript. Please find the detailed response to each of your comments below.**

1. Dr Heinze presents a local thermal non-equilibrium model for infiltration of water in snow. The motivation is to develop a numerical model to consider the thermal energy related to melting of ice or freezing of liquid water through the snowpack. The non-equilibrium model is interesting. However, it would be worthwhile to compare results from the same thermo-hydraulic scenarios with results from an equilibrium model. It would be useful to evaluate if there are conditions when a simpler equilibrium model is adequate.

   **Reply: Thank you very much for pointing me to this lack in the analysis and the obvious reader's interest in the answer to this question. I will include a comparison to an equilibrium model and discuss the potential relevance of non-equilibrium models in dependence of snow properties and boundary conditions.**

2. The coupling with the hydraulic conductivity of the snow is rudimentary in that thermo-hydraulic processes are investigated for influxes of water into snowpacks consisting of spherical grains. The author acknowledges that this is not realistic, but this condition could apply during infiltration through 'ripe' snow that has previously been wetted and subjected to grain growth (Colbeck, 1979; Raymond and Tusima, 1979).

   **Reply: Thank you very much for pointing me to the condition of ripe snow. I will include this reference and change the text accordingly.**

3. However, natural snowpacks are typically layered and heterogeneous; during infiltration, the snow structure and density, and flow fingering often evolve rapidly (e.g. Colbeck, 1979; Marshall et al. 1999; Marshall et al., 2014, Hirashima et al., 2017; Katsushima, 2020; Ohara, 2024). Forecasting impacts of ROS on flash floods and snow avalanches requires modeling thermo-hydraulic processes in natural snowpacks.

   **Reply: I fully agree that the presented model is only a first step into applying LTNE models for simulating natural ROS events and related hazards. The focus of this work is to build the mathematical ground work and to develop the physical concepts. Hence, future extensions of the model towards two-dimensions to account for horizontal heterogeneity or even for dual-domain approaches (for rock: Heinze & Hamidi, 2017) are clearly envisioned. For showcasing the strength and ability of the model, the field observations from 15/16th January 1992 presented in Conway & Benedict (1994) will be numerically reproduced using the presented model. Such a simulation result and its discussion will be added to the manuscript. This will demonstrate the ability of the model to account for one-dimensional heterogeneity.**

4. Dr Heinze mentions that different snow morphology and layering also need to be considered; you might be interested in a study using a water transport model, a dual-domain approach and a multi-layer SNOWPACK model to study infiltration of water in a layered snowpack. (Hirashima et al., 2018)

   **Reply: Thank you very much for pointing me to this very interesting study. Combining the presented thermal non-equilibrium model with more realistic representations of snow hydrology and morphology, also in the context of a dual-**

**domain approach, are surely necessary for the simulation of realistic events. The reference will be added with a small extension of the discussion into the manuscript.**

**References:**

**Conway, H., and R. Benedict, 1994. Infiltration of water into snow. Water Resources Research 30(3), 641-649. https://doi.org/10.1029/93WR03247**

**Heinze, T., and S. Hamidi, 2017: Heat transfer and parameterization in local thermal non-equilibrium for dual porosity continua. Applied Thermal Engineering 114, 645-652.**

Reviewer's Reverences:

Colbeck, SC 1979.Water flow through heterogeneous snow. Cold Regions Sci and Tech/. 1, 37-45

Hirashima, H., F. Avanzi, and S. Yamaguchi, 2017: Liquid water infiltration into a layered snowpack: evaluation of a 3D water transport model with laboratory experiments, Hydrol. Earth Syst. Sci. Discuss., 2017, 1–22, doi:10.5194/hess-2017-200.

Hiroyuki Hirashima, Nander Wever, Francesco Avanzi, Satoru Yamaguchi, Yoshiyuki Ishii 2016. Simulating liquid water infiltration – comparison between a three-dimensional water transport model and a dual-domain approach using snowpack. Proceedings, International Snow Science Workshop, Innsbruck, Austria, 2018

Takafumi Katsushima, Satoru Adachi, Satoru Yamaguchi, Toshihiro Ozeki,Toshiro Kumakura, 2020. Nondestructive three-dimensional observations of flow finger and lateral low development in dry snow using magnetic resonance imaging. Cold Regions Science and Technology 170 (2020) 102956

Marshall, HP, H Conway, LA Rasmussen, 1999: Snow densification during rain Cold Regions Science and Technology 30 1999. 35–41

Marshall, HP and the Cryosphere Geophysics and Remote Sensing (CryoGARS) group, 2014. Water in snow likes to go with the flow: dynamics of liquid water in snow and its impact on stability. Proceedings, International Snow Science Workshop, Banff, 2014

Noriaki Ohara, 2024. Finger flow modeling in snow porous media based on lagrangian mechanics. Advances in Water Resources 185 (2024) 104634

Raymond, CF, Tusima, K, 1979. Grain coarsening of water-saturated snow. J. Glaciology, Vol.22 No.86,1979

---

## Author Comment (AC2)

Response to Reviewer Comments #2 (**Responses in bold**)

**Thank you for taking the time to conduct this thorough review and for your constructive comments, which will be used to substantially improve the manuscript. Please find the detailed response to each of your comments below.**

The author presents an up-to-date theoretical model of meltwater infiltration in snow and soil assuming a thermal non-equilibrium (TNE) between the vapor, water and ice phases. The modelling approach, tested in numerical experiments, is novel and interesting but experimental evidences to support the theoretical assumptions and the improvements in the modelling approach are missing. They would provide a significant added value to the research. Otherwise it is not very clear which is the added value of the modelling approach in terms of simulating the actual water and heat dynamics into the snowpack and the frozen soil. Therefore, if experimental data are not provided to support the model's hypothesis at least some simulations under the hypothesis of thermal equilibrium (TE), showing the differences between the TNE and TE assumptions, and simplified traditional hypotheses of advective heat transfer available for melt M (melt rate) as M=PT/80, with P being the Rain on snow intensity, T the air temperature and 80 the ratio of specific heat capacity of water and latent heat of fusion, are recommended. In this way the improvements introduced by the model would be more evident.

**Reply: Finding a suitable experimental data set for quantitative comparison with the numerical model is difficult due to the current limitations of the model (1D, no preferential pathways) and its required input data (hydraulic parameters, thermal boundary conditions, etc.). Generally, measuring separate phase temperatures in snow seems experimentally challenging. Still, to showcase the strength and ability of the model, the field observations from 15/16$^{th}$ January 1992 presented in Conway & Benedict (1994) can be numerically reproduced using the presented model within the before mentioned limits and assuming a suitable set of parameters. Such a simulation will be added to the manuscript. This will also demonstrate the ability of the model to account for one-dimensional heterogeneity. This dataset has been selected due to its well-documented rainfall conditions and the thermal as well as hydraulic propagation within the snow allowing to match needed parameters accordingly. Also, the relevant snow types are described as partly rounded or rounded grains fitting the theoretical assumption of the model. For future work, there is hope that with the presented work, further experimental research addressing potential thermal non-equilibrium situations in snow might be initiated and can subsequently be used to further constrain the model.**

**In general, the benefits of the presented model compared to conventional approaches are an improved process-understanding of the thermo-hydraulic processes within the snowpack, the possibility to investigate the influential parameters for local freezing/melting conditions and a consistent mathematical formulation of boundary conditions without the a-priori simplification of thermal equilibrium of all phases. I believe that challenging the simplifying assumption of instant thermal equilibrium between phases for rain-on-snow events is a valuable approach in itself based on the initial thermal non-equilibrium condition. As suggested, a comparison of the newly presented model to a thermal equilibrium model will be added to the manuscript to investigate relevant conditions of LTE/LTNE, which will help to clarify when the**

**explicit description of heat transfer is essential for the processes within the snowpack and when it can be omitted.**

**Please note that the proposed traditional method of M=PT/80 is assuming that the snow is close to melting point already. In the presented simulations that is not necessarily the case, so attention needs to be paid to apply this method only for the melting stage. Nevertheless, comparing model results and the analytical method will demonstrate the agreement of the novel model and traditional approaches for relevant macroscopic quantities and will be added to the manuscript.**

Some key references are missing as suggested in the review.

Line 30 Literature in the 70s and 80s posed the bases for multiphase snowpack dynamics and meltwater infiltration into snow. I added some fundamental references (Colbeck, 1972, 1978: Colbeck and Anderson, 1982; Dunne et al., 1976; Morris, 1991; Akan, 1984a, 1984b) that cannot be neglected, also in view of the model's parameterization and verification with experimental data.
Line 35 About soil freezing and thawing I would refer also to Leuther and Schlüter (2021)

**Reply: Thank you very much for pointing me to this relevant literature, which will be included in the literature review and added to the reference list.**

Line 52 I would spell out LTE Local Thermal Equilibrium

**Reply: Thank you very much for pointing me to this lack of introducing the abbreviation. Of course, this will be changed.**

Line 75 I suggest to give some more references about the capacity of the van Genuchten model (developed for soils) to explain water saturation-hydraulic head relationship also for snow.

**Reply: Thank you very much for pointing out this lack of references for a critical component of the hydraulic model. The references (Jordan et al., 1995; Yamaguchi et al., 2010; Yamaguchi et al., 2017) will be added to the manuscript and shortly discussed.**

Line 125 Explain better the assumption about a similar flow velocity for air and infiltrating water. Water is forced by gravity and capillary forces that cannot be treated in the same way for air.

**Reply: This simplifying assumption of equal flow velocities is a consequence of other assumptions made, such as the incompressibility of water and air, the capillary tube model, and the exclusion of mixture flow within one capillary tube. Hence, if water replaces air during infiltration, conservation of mass requires the same flow velocity if**

the tube diameter does not change. Naturally, in a 3D reality the flow paths of the air are complex and through various capillary tubes which cannot be represented here. However, due to the negligible thermal influence of the air, this simplifying assumption has no impact on the simulations' outcome. Based on your comment, the explanation of the assumption will be extended.

Line 130 Specify the meaning of subscript ij (the 3 phases of water?) for Qij, hij and Aij

Reply: Thank you for pointing out the missing explanation. The subscripts ij indicate the involved phases, which exchange heat. Hence, ij represent the possible heat transfer combinations solid-liquid (sw), solid-air (sa), liquid-air (wa). The respective explanation will be added to the text.

Line 212 In Table 1. Ice density is assumed 917 kg/m3 a value generally adopted in the literature. Why is ice density assumed 940 kg/m3 at line 212?

Reply: This is indeed a mistake, which will be corrected. Throughout the manuscript, the ice density has been set to 917 kg/m^3 and used accordingly.

Line 212 The assumption of a spherical shape for snow crystals with low density as 0.1 kg/m3 is not very realistic as for that density a dendritic shape of snow crystals is more appropriate. Which are the implications of this assumption for the model proposed?

Reply: The assumption of spherical snow grains is obviously a strong limitation of the model but enables a consistent mathematical formulation also accounting for growth and decline of snow grain diameter. The spherical shape is used in the model to calculate the surface area of the snow for the heat exchange terms. The linear dependence of this is shown in equ. 13. Hence, more surface area increases the heat transfer across that surface. Also, estimations of the infiltration behavior (vanGenuchten parameters) taken from literature consider spherical grains. As these are empirical equations, implications on the hydraulic side are difficult to assess. Based on your comment, a respective explanation will be included in the respective paragraph.

Line 230 The explanation of the mechanical compaction of snow needs to be better explained.

Reply: The mechanical compaction is partly based on the weight of the snow and the rainwater infiltrating but there are also changes to the crystal and grain structure (Marshall et al. 1999). Melting might occur, as seen in the simulation result discussed here, in deeper layers of the snowpack and not necessarily at the surface. Hence, changes in the snow pack structure might cause collapse due to the load above. An extended explanation will be added to the text and the references (Bertle et al., 1966; Marshall et al., 1999; Meyer & Hewitt, 2017; Barraclough et al., 2017) will be included

**in the manuscript. Please note that the model itself does not account for mechanical compaction.**

Line 250 A rainfall depth of 0.1 m is assumed but over which time period does rainfall occur? Then it seems that in the modelling approach a constant hydraulic head of 0.1 m holds at the top boundary (see figure 1, 2, 3, 4, 5, 6, 7). Is this the head of a constant water depth (totally unrealistic) or does it include the capillary head?

**Reply: The boundary condition of 0.1m constant water head was chosen in analogy to laboratory experiments on frozen soils (e.g. Hansson et al., 2004). This constant pressure head boundary condition leads to a non-linear infiltration pattern into the unsaturated snow which might not be fully representative for a natural rain event and might lead to infiltration rates of more than 40mm/hour, which are comparably high for rain-on-snow events (cf. Juras et al., 2021) but were used in rain-on-snow experiments (cf. Yang et al., 2023). The investigation of varying rainfall intensities is an interesting point, that will be addressed in the manuscript by adding additional simulations with varying boundary conditions changing the top boundary condition from a constant head to an infiltration condition. A smaller rainfall intensity leads to smaller amounts of rainwater entering the snowpack delaying warming of the snow and melting.**

**Also based on this comment, the description of the top boundary conditions will be clarified in the text.**

Line 254-292 This numerical simulation is interesting. But how would the melt

**Reply: Sadly, your comment was abbreviated in your review.**

Discussion. Some discussion about perspectives of the modelling approach to test its results for instance testing its results with measurements of snowpack properties and passive microwave monitoring of the freezing/melting processes as in Cagnati et al. (2004) would be useful.

**Reply: Thank you very much for this comment and pointing me to the respective references. Such a data source, ideally at a high spatial resolution of 5 cm or less, would be greatly valuable to further constrain the heat transfer processes in the model. However, for comparison with long-time monitoring data, the model would probably also need to include more processes on the boundaries as well as internally (compaction, snow metamorphosis). Based on your comment, a systematic discussion of arising possibilities to compare the model with respective monitoring techniques will be added to the manuscript.**

How would the infiltration fluxes change if a hydraulic head of 0.001 m is assumed at the top boundary? The top boundary hydraulic head conditions are not very clear (see comment to line 250).

**Reply: Please also see my reply to your comment above. This is an interesting point and will be addressed by adding simulations with a top boundary condition varying rainfall intensity to study the effect of different precipitation events.**

Line 408. If experimental data are not provided to support the model's hypothesis at least some simulations under the hypothesis of thermal equilibrium (TE), showing the differences between the TNE and TE assumptions, and simplified traditional hypotheses of advective heat transfer would be useful.

**Reply: Please see my reply to your earlier comment regarding this suggestion. A comparison to field data, to a thermal equilibrium model and to a traditional analytical model, as suggested by you above, will be added to the manuscript.**

**References**

**Barraclough, T., Blackford, J., Liebenstein, S. et al. Propagating compaction bands in confined compression of snow. Nature Phys 13, 272–275 (2017). https://doi.org/10.1038/nphys3966**

**Bertle, F.A., 1966. Effect of snow compaction on runoff from Rain on Snow. A Water Resources Technical Publication Engineering Monography 35. United States Department of the Interior.**

**Conway, H., and R. Benedict, 1994. Infiltration of water into snow. Water Resources Research 30(3), 641-649. https://doi.org/10.1029/93WR03247**

**Hansson, K., Simunek, J., Mizoguchi, M. et al., 2004. Water Flow and Heat Transport in Frozen Soil: Numerical Solution and Freeze-Thaw Applications. Vade Zone Journal 3, 693-704.**

**Jordan, R., 1995. Effects of Capillary Discontinuities on Water Flow and Water Retention in Layered Snowcovers. Defence Science Journal 45(2), 79-91.**

**Juras, R., Blöcher, J.R., Jenicek, M., Hotovy, O., Markonis, Y., 2021. What affects the hydrological response of rain-on-snow events in low-altitude mountain ranges in Central Europe?, Journal of Hydrology 603, https://doi.org/10.1016/j.jhydrol.2021.127002.**

**Leroux, N.R., Marsh, C.B, and J.W. Pomeroy, 2020: Simulation of Preferential Flow in Snow with a 2-D Non-Equilibrium Richards Model and Evaluation Against Laboratory Data. Water Resources Research 56, e2020WR027466.**

**Marshall, H.P., Conway, H., and L.A. Rasmussen, 1999. Snow densification during rain. Cold Region Science and Technology 30(1-3), 35-41.**

**Meyer, C.R., and I.J. Hewitt, 2017. A continuum model for meltwater flow through compacting snow. The Cryosphere 11, 2799-2813. doi.org/10.5194/tc-11-2799-2017.**

**Yamaguchi, S., Watanabe, K., Katsushima, T., et al., 2012. Dependence of the water retention curve of snow on snow characteristics. Annals of Glaciology 53(61), 6-12.**

**Yamaguchi, S., Katsushima, T., Sato, A., et al., 2010. Water retention curve of snow with different grain sizes. Cold Regions Science and Technology 64, 87-93.**

**Yang, Z., Chen, R., Liu, Y., Zhao, Y., Liu, Z., & Liu, J. (2023). The impact of rain-on-snow events on the snowmelt process: A field study. Hydrological Processes, 37(11), e15019. https://doi.org/10.1002/hyp.15019**

Reviewer's references

Akan, A. O.: 1984a, 'Mathematical Simulation of Snowmelt and Runoff from Snow Covers', *Frontiers in Hydrology*, Water Resources Publications, pp. 79–92. https://doi-org.proxy.unibs.it/10.1029/WR020i006p00707

Akan, A. O. (1984b). Simulation of runoff from snow-covered hillslopes. *Water Resources Research*, *20*(6), 707-713.

Cagnati, A., A. Crepaz, G. Macelloni, P. Pampaloni, R. Ranzi, M. Tedesco, M. Tomirotti and M. Valt, Study of the snow melt–freeze cycle using multi–sensor data and snow modelling, J. of Glaciology, 50(170), 419-426, 2004

Colbeck, S. C.: 1972, 'A Theory of Water Percolation in Snow,' *Journal of Glaciology* 11(63), 369–385.

Colbeck, S. C. (1978). The physical aspects of water flow through snow. In *Advances in hydroscience* (Vol. 11, pp. 165-206). Elsevier.

Colbeck, S. C., and E. A. Anderson: 1982, 'The Permeability of a Melting Snow Cover,' *Water Resources Research* 18(4), 904–908.

https://doi-org.proxy.unibs.it/10.1029/WR018i004p00904

 Dunne, T., Price, A. G., & Colbeck, S. C. (1976). The generation of runoff from subarctic snowpacks. *Water Resources Research*, *12*(4), 677-685.

https://doi.org/10.1029/WR012i004p00677

Morris, E.M. (1991). Physics-Based Models of Snow. In: Bowles, D.S., O'Connell, P.E. (eds) Recent Advances in the Modeling of Hydrologic Systems. NATO ASI Series, vol 345. Springer, Dordrecht. https://doi.org/10.1007/978-94-011-3480-4_5

Leuther, F. and Schlüter, S.: Impact of freeze–thaw cycles on soil structure and soil hydraulic properties, SOIL, 7, 179–191, https://doi.org/10.5194/soil-7-179-2021, 2021.

---

## Author Response (AR1)

Response to Reviewer Comments #1 (**Responses in bold**)

**Thank you for taking the time to conduct this thorough review and for your constructive comments, which has been used to substantially improve the manuscript. Please find the detailed response to each of your comments below. Referenced lines refer to the manuscript with marked changes.**

1. Dr Heinze presents a local thermal non-equilibrium model for infiltration of water in snow. The motivation is to develop a numerical model to consider the thermal energy related to melting of ice or freezing of liquid water through the snowpack. The non-equilibrium model is interesting. However, it would be worthwhile to compare results from the same thermo-hydraulic scenarios with results from an equilibrium model. It would be useful to evaluate if there are conditions when a simpler equilibrium model is adequate.

   **Reply: Thank you very much for pointing me to this lack in the analysis and the obvious reader's interest in the answer to this question.**

   **I added the simulation of the field observation by 15/16th January 1992 presented in Conway & Benedict (1994) to the manuscript in section 3.1 because**

   **(1) The heterogeneous snowpack described by Conway & Benedict (1994) allows to showcase the model's ability to incorporate that.**

   **(2) The comparison to actual field data, a thermal equilibrium model of the same event and a simple analytical approach strengthens the trust into the model and the simulation results while at the same time demonstrates the improvement of the newly developed model compared to conventional approaches.**

   **It becomes evident that a warmed snowpack (close to 0°C) with no thermal gradient can be described just fine with a simpler equilibrium model. However, in the presence of thermal gradients within the snowpack, especially if layering hinders water infiltration, the developed model provides an improved representation of the thermos-hydraulic state of the snowpack.**

   **For the respective changes please see the added subsection 3.1 (starting l. 285) with three new figures (Fig. 1-3), as well as the revised Discussion (starting l. 525) and the new Conclusion section.**

2. The coupling with the hydraulic conductivity of the snow is rudimentary in that thermo-hydraulic processes are investigated for influxes of water into snowpacks consisting of spherical grains. The author acknowledges that this is not realistic, but this condition could apply during infiltration through 'ripe' snow that has previously been wetted and subjected to grain growth (Colbeck, 1979; Raymond and Tusima, 1979).

   **Reply: Thank you very much for pointing me to the condition of ripe snow. The discussion and the respective references have been added in section 2 in lines 243-248.**

3. However, natural snowpacks are typically layered and heterogeneous; during infiltration, the snow structure and density, and flow fingering often evolve rapidly (e.g. Colbeck, 1979; Marshall et al. 1999; Marshall et al., 2014, Hirashima et al., 2017; Katsushima, 2020; Ohara, 2024). Forecasting impacts of ROS on flash floods and snow avalanches requires modeling thermo-hydraulic processes in natural snowpacks.

**Reply: I fully agree that the presented model is only a first step into applying LTNE models for simulating natural ROS events and related hazards. The focus of this work is to build the mathematical ground work and to develop the physical concepts. Hence, future extensions of the model towards two-dimensions to account for horizontal heterogeneity or even for dual-domain approaches (for rock: Heinze & Hamidi, 2017) are clearly envisioned. Please see my reply to comment #1 regarding the current model's ability to account for 1D heterogeneity.**

**Based on your comment, I extended the respective discussion of this important feature. Please see section 4.4 starting l. 587.**

4. Dr Heinze mentions that different snow morphology and layering also need to be considered; you might be interested in a study using a water transport model, a dual-domain approach and a multi-layer SNOWPACK model to study infiltration of water in a layered snowpack. (Hirashima et al., 2018)

**Reply: Thank you very much for pointing me to this very interesting study. Combining the presented thermal non-equilibrium model with more realistic representations of snow hydrology and morphology, also in the context of a dual-domain approach, are surely necessary for the simulation of realistic events.**

**Please see my response above to the respective changes in the manuscript.**

**References:**

**Conway, H., and R. Benedict, 1994. Infiltration of water into snow. Water Resources Research 30(3), 641-649. https://doi.org/10.1029/93WR03247**

**Heinze, T., and S. Hamidi, 2017: Heat transfer and parameterization in local thermal non-equilibrium for dual porosity continua. Applied Thermal Engineering 114, 645-652.**

Reviewer's Reverences:

Colbeck, SC 1979.Water flow through heterogeneous snow. Cold Regions Sci and Tech/. 1, 37-45

Hirashima, H., F. Avanzi, and S. Yamaguchi, 2017: Liquid water infiltration into a layered snowpack: evaluation of a 3D water transport model with laboratory experiments, Hydrol. Earth Syst. Sci. Discuss., 2017, 1–22, doi:10.5194/hess-2017-200.

Hiroyuki Hirashima, Nander Wever, Francesco Avanzi, Satoru Yamaguchi, Yoshiyuki Ishii 2016. Simulating liquid water infiltration – comparison between a three-dimensional water transport model and a dual-domain approach using snowpack. Proceedings, International Snow Science Workshop, Innsbruck, Austria, 2018

Takafumi Katsushima, Satoru Adachi, Satoru Yamaguchi, Toshihiro Ozeki,Toshiro Kumakura, 2020. Nondestructive three-dimensional observations of flow finger and lateral low development in dry snow using magnetic resonance imaging. Cold Regions Science and Technology 170 (2020) 102956

Marshall, HP, H Conway, LA Rasmussen, 1999: Snow densification during rain Cold Regions Science and Technology 30 1999. 35–41

Marshall, HP and the Cryosphere Geophysics and Remote Sensing (CryoGARS) group, 2014. Water in snow likes to go with the flow: dynamics of liquid water in snow and its impact on stability. Proceedings, International Snow Science Workshop, Banff, 2014

Noriaki Ohara, 2024. Finger flow modeling in snow porous media based on lagrangian mechanics. Advances in Water Resources 185 (2024) 104634

Raymond, CF, Tusima, K, 1979. Grain coarsening of water-saturated snow. J. Glaciology, Vol.22 No.86,1979

Response to Reviewer Comments #2 (**Responses in bold**)

**Thank you for taking the time to conduct this thorough review and for your constructive comments, which have been used to substantially improve the manuscript. Please find the detailed response to each of your comments below. Referenced lines refer to the manuscript with marked changes.**

The author presents an up-to-date theoretical model of meltwater infiltration in snow and soil assuming a thermal non-equilibrium (TNE) between the vapor, water and ice phases. The modelling approach, tested in numerical experiments, is novel and interesting but experimental evidences to support the theoretical assumptions and the improvements in the modelling approach are missing. They would provide a significant added value to the research. Otherwise it is not very clear which is the added value of the modelling approach in terms of simulating the actual water and heat dynamics into the snowpack and the frozen soil. Therefore, if experimental data are not provided to support the model's hypothesis at least some simulations under the hypothesis of thermal equilibrium (TE), showing the differences between the TNE and TE assumptions, and simplified traditional hypotheses of advective heat transfer available for melt M (melt rate) as M=PT/80, with P being the Rain on snow intensity, T the air temperature and 80 the ratio of specific heat capacity of water and latent heat of fusion, are recommended. In this way the improvements introduced by the model would be more evident.

**Reply: Finding a suitable experimental data set for quantitative comparison with the numerical model is difficult due to the current limitations of the model (1D, no preferential pathways) and its required input data (hydraulic parameters, thermal boundary conditions, etc.). Generally, measuring separate phase temperatures in snow seems experimentally challenging.**
**To showcase the strength and ability of the model, the field observations from 15/16[th] January 1992 presented in Conway & Benedict (1994) are numerically reproduced using the presented model and the results were added to the Results section as subsection 3.1. This dataset has been selected due to its well-documented rainfall conditions and the thermal as well as hydraulic propagation within the snow allowing to match needed parameters accordingly. Also, the relevant snow types are described as partly rounded or rounded grains fitting the theoretical assumption of the model.**
**The same field observation has been used to compare the model results with a thermal equilibrium model, the simple analytical approach you presented above and to study the effect of rainfall intensity (see your comment below).**

**Altogether, the simulation of this specific ROS events showcases the model's ability to incorporate 1D heterogeneity of the snowpack, strengthens the trust into the model and the simulation results while at the same time demonstrates the improvement of the newly developed model compared to conventional approaches.**

**It becomes evident that a warmed snowpack (close to 0°C) with no thermal gradient can be described just fine with a simpler equilibrium model. However, in the presence of thermal gradients within the snowpack, especially if layering hinders water infiltration, the developed model provides an improved representation of the thermos-hydraulic state of the snowpack. Please see the new results section 3.1 (starting l. 285), the**

**restructured discussion section (starting l. 525), as well as the newly added Conclusion section (starting l. 624).**

Some key references are missing as suggested in the review.

Line 30 Literature in the 70s and 80s posed the bases for multiphase snowpack dynamics and meltwater infiltration into snow. I added some fundamental references (Colbeck, 1972, 1978: Colbeck and Anderson, 1982; Dunne et al., 1976; Morris, 1991; Akan, 1984a, 1984b) that cannot be neglected, also in view of the model's parameterization and verification with experimental data.
Line 35 About soil freezing and thawing I would refer also to Leuther and Schlüter (2021)

**Reply: Thank you very much for pointing me to this relevant literature, which has been included in the literature review. Please see lines 29-37 & 44-45 in the Introduction.**

Line 52 I would spell out LTE Local Thermal Equilibrium

**Reply: Thank you very much for pointing me to this lack of introducing the abbreviation. Of course, this has been changed. Please see line 62.**

Line 75 I suggest to give some more references about the capacity of the van Genuchten model (developed for soils) to explain water saturation-hydraulic head relationship also for snow.

**Reply: Thank you very much for pointing out this lack of references for a critical component of the hydraulic model. Please see the extended discussion in lines 87-96.**

Line 125 Explain better the assumption about a similar flow velocity for air and infiltrating water. Water is forced by gravity and capillary forces that cannot be treated in the same way for air.

**Reply: This simplifying assumption of equal flow velocities is a consequence of other assumptions made, such as the incompressibility of water and air, the capillary tube model, and the exclusion of mixture flow within one capillary tube. Hence, if water replaces air during infiltration, conservation of mass requires the same flow velocity if the tube diameter does not change. Naturally, in a 3D reality the flow paths of the air are complex and through various capillary tubes which cannot be represented here. However, due to the negligible thermal influence of the air, this simplifying assumption has no impact on the simulations' outcome.**
**Based on your comment, the explanation has been extended. Please see lines 145-150.**

Line 130 Specify the meaning of subscript ij (the 3 phases of water?) for Qij, hij and Aij

**Reply: Thank you for pointing out the missing explanation. The subscripts ij indicate the involved phases, which exchange heat. The respective explanation has been added to the text and usage of subscripts was checked and modified again for consistency. Please see lines 154-156.**

Line 212 In Table 1. Ice density is assumed 917 kg/m3 a value generally adopted in the literature. Why is ice density assumed 940 kg/m3 at line 212?

**Reply: This is indeed a mistake, which has been corrected. Throughout the manuscript, the ice density has been set to 917 kg/m^3 and used accordingly.**

Line 212 The assumption of a spherical shape for snow crystals with low density as 0.1 kg/m3 is not very realistic as for that density a dendritic shape of snow crystals is more appropriate. Which are the implications of this assumption for the model proposed?

**Reply: The assumption of spherical snow grains is obviously a strong limitation of the model but enables a consistent mathematical formulation also accounting for growth and decline of snow grain diameter. The spherical shape is used in the model to calculate the surface area of the snow for the heat exchange terms. The linear dependence of this is shown in equ. 13. Hence, more surface area increases the heat transfer across that surface. Also, estimations of the infiltration behavior (vanGenuchten parameters) taken from literature consider spherical grains. As these are empirical equations, implications on the hydraulic side are difficult to assess.**

**Based on your comment, a respective explanation has been added. Please note a mistake in the original manuscript. Snow densities considered are 100 – 800 kg/m^3 not 0.1 – 0.8. Please see lines 242-249 for the changes.**

Line 230 The explanation of the mechanical compaction of snow needs to be better explained.

**Reply: The mechanical compaction is partly based on the weight of the snow and the rainwater infiltrating but there are also changes to the crystal and grain structure (Marshall et al. 1999). Melting might occur, as seen in the simulation result discussed here, in deeper layers of the snowpack and not necessarily at the surface. Hence, changes in the snow pack structure might cause collapse due to the load above. An extended explanation has been added to the manuscript. Please see lines 264-270.**

Line 250 A rainfall depth of 0.1 m is assumed but over which time period does rainfall occur? Then it seems that in the modelling approach a constant hydraulic head of 0.1 m holds at the top boundary (see figure 1, 2, 3, 4, 5, 6, 7). Is this the head of a constant water depth (totally unrealistic) or does it include the capillary head?

**Reply: The boundary condition of 0.1m constant water head was chosen in analogy to laboratory experiments on frozen soils (e.g. Hansson et al., 2004). This constant pressure head boundary condition leads to a non-linear infiltration pattern into the unsaturated snow which might not be fully representative for a natural rain event and might lead to infiltration rates of more than 40mm/hour, which are comparably high for rain-on-snow events (cf. Juras et al., 2021) but were used in rain-on-snow experiments (cf. Yang et al., 2023).**
**Based on your comment, the investigation of varying rainfall intensities has been added to the manuscript in the results section 3.1.**
**Also based on this comment, the description of the top boundary conditions has been clarified in the text. Please see lines 274-277 & 372-373.**

Line 254-292 This numerical simulation is interesting. But how would the melt

**Reply: Sadly, your comment was abbreviated in your review.**

Discussion. Some discussion about perspectives of the modelling approach to test its results for instance testing its results with measurements of snowpack properties and passive microwave monitoring of the freezing/melting processes as in Cagnati et al. (2004) would be useful.

**Reply: Thank you very much for this comment and pointing me to the respective references. Such a data source, ideally at a high spatial resolution of 5 cm or less, would be greatly valuable to further constrain the heat transfer processes in the model. However, for comparison with long-time monitoring data, the model would probably also need to include more processes on the boundaries as well as internally (compaction, snow metamorphosis). Based on your comment, a systematic discussion of arising possibilities to compare the model with respective monitoring techniques has been added to the manuscript. Please see lines 572 – 586 in the new respective subsection in the discussion.**

How would the infiltration fluxes change if a hydraulic head of 0.001 m is assumed at the top boundary? The top boundary hydraulic head conditions are not very clear (see comment to line 250).

**Reply: Please also see my reply to your comment above. The effect of varying rainfall intensities has been added to the Results section 3.1.**

Line 408. If experimental data are not provided to support the model's hypothesis at least some simulations under the hypothesis of thermal equilibrium (TE), showing the differences between the TNE and TE assumptions, and simplified traditional hypotheses of advective heat transfer would be useful.

**Reply: Please see my reply to your earlier comment regarding this suggestion. A comparison to field data, to a thermal equilibrium model and to a traditional analytical model, as suggested by you above, has been added to the manuscript in results section 3.1.**

**References**

Barraclough, T., Blackford, J., Liebenstein, S. et al. Propagating compaction bands in confined compression of snow. Nature Phys 13, 272–275 (2017). https://doi.org/10.1038/nphys3966

Bertle, F.A., 1966. Effect of snow compaction on runoff from Rain on Snow. A Water Resources Technical Publication Engineering Monography 35. United States Department of the Interior.

Conway, H., and R. Benedict, 1994. Infiltration of water into snow. Water Resources Research 30(3), 641-649. https://doi.org/10.1029/93WR03247

Hansson, K., Simunek, J., Mizoguchi, M. et al., 2004. Water Flow and Heat Transport in Frozen Soil: Numerical Solution and Freeze-Thaw Applications. Vade Zone Journal 3, 693-704.

Jordan, R., 1995. Effects of Capillary Discontinuities on Water Flow and Water Retention in Layered Snowcovers. Defence Science Journal 45(2), 79-91.

Juras, R., Blöcher, J.R., Jenicek, M., Hotovy, O., Markonis, Y., 2021. What affects the hydrological response of rain-on-snow events in low-altitude mountain ranges in Central Europe?, Journal of Hydrology 603, https://doi.org/10.1016/j.jhydrol.2021.127002.

Leroux, N.R., Marsh, C.B, and J.W. Pomeroy, 2020: Simulation of Preferential Flow in Snow with a 2-D Non-Equilibrium Richards Model and Evaluation Against Laboratory Data. Water Resources Research 56, e2020WR027466.

Marshall, H.P., Conway, H., and L.A. Rasmussen, 1999. Snow densification during rain. Cold Region Science and Technology 30(1-3), 35-41.

Meyer, C.R., and I.J. Hewitt, 2017. A continuum model for meltwater flow through compacting snow. The Cryosphere 11, 2799-2813. doi.org/10.5194/tc-11-2799-2017.

Yamaguchi, S., Watanabe, K., Katsushima, T., et al., 2012. Dependence of the water retention curve of snow on snow characteristics. Annals of Glaciology 53(61), 6-12.

Yamaguchi, S., Katsushima, T., Sato, A., et al., 2010. Water retention curve of snow with different grain sizes. Cold Regions Science and Technology 64, 87-93.

Yang, Z., Chen, R., Liu, Y., Zhao, Y., Liu, Z., & Liu, J. (2023). The impact of rain-on-snow events on the snowmelt process: A field study. Hydrological Processes, 37(11), e15019. https://doi.org/10.1002/hyp.15019

Reviewer's references

Akan, A. O.: 1984a, 'Mathematical Simulation of Snowmelt and Runoff from Snow Covers', *Frontiers in Hydrology*, Water Resources Publications, pp. 79–92. https://doi-org.proxy.unibs.it/10.1029/WR020i006p00707

Akan, A. O. (1984b). Simulation of runoff from snow-covered hillslopes. *Water Resources Research*, *20*(6), 707-713.

Cagnati, A., A. Crepaz, G. Macelloni, P. Pampaloni, R. Ranzi, M. Tedesco, M. Tomirotti and M. Valt, Study of the snow melt–freeze cycle using multi–sensor data and snow modelling, J. of Glaciology, 50(170), 419-426, 2004

Colbeck, S. C.: 1972, 'A Theory of Water Percolation in Snow,' *Journal of Glaciology* 11(63), 369–385.

Colbeck, S. C. (1978). The physical aspects of water flow through snow. In *Advances in hydroscience* (Vol. 11, pp. 165-206). Elsevier.

Colbeck, S. C., and E. A. Anderson: 1982, 'The Permeability of a Melting Snow Cover,' *Water Resources Research* 18(4), 904–908.

https://doi-org.proxy.unibs.it/10.1029/WR018i004p00904

Dunne, T., Price, A. G., & Colbeck, S. C. (1976). The generation of runoff from subarctic snowpacks. *Water Resources Research*, *12*(4), 677-685.

https://doi.org/10.1029/WR012i004p00677

Morris, E.M. (1991). Physics-Based Models of Snow. In: Bowles, D.S., O'Connell, P.E. (eds) Recent Advances in the Modeling of Hydrologic Systems. NATO ASI Series, vol 345. Springer, Dordrecht. https://doi.org/10.1007/978-94-011-3480-4_5

Leuther, F. and Schlüter, S.: Impact of freeze–thaw cycles on soil structure and soil hydraulic properties, SOIL, 7, 179–191, https://doi.org/10.5194/soil-7-179-2021, 2021.

---

## Editor Decision (ED1)

Line numbers refer to the version with tracked changes.

1) Lines 81 & 82. Sentence should be rephrased: "it potentially affects" is not clearly related to the preceding part of the sentence (where the subject is plural, namely "LTNE effects").
2) Lines 88 & 89. Sentence "Following studies varying meteorological and thermo-hydraulic parameters further reveal conditions  promoting LTNE effects and address the uncertainty in model parameterization" should be rephrased.
3) Line 95. I prefer "matric potential" to "pressure head".
4) Line 98. It would be useful to recall the measurement units for $\alpha$, i.e., [m$^{-1}$].
5) Equation (3). I am afraid that this equation is incorrect. If I am not missing something, it does not correspond to equation (23) of van Genuchten (1980). Basically, $(\epsilon_{l,sat} - \epsilon_{l,ref})^{1-m}$ should be substituted with $(\epsilon_{l,sat} - \epsilon_{l,ref}) \times (1 - m)^{-1}$. Am I wrong?
6) Lines 103, 104, 107, 108, 259. A space Is missing in "van Genuchten".
7) Line 108. A space should be added after "season".
8) Line 111. Word "been" is missing in "have also successfully used".
9) Equation (4). It is necessary to recall the measurement units for $\alpha$. In particular the equation could be rewritten as $\alpha$ = 7.3 m$^{-1}$ × mm$^{-1}$ × $d$ + 1.9 m$^{-1}$.
10) Equation (5). Analogously to (4), a preferred format for this equation is n = -3.3 mm × $d$ + 14.4.
11) Line 127. I think that "From rocks" should be substituted, possibly with "For saturated porous media".
12) Equation (7). The definition of phi and phi0 is missing.
13) Line 142. Sentence "Throughout... respectively" should appear much early, when the indices for the three phases are used for the first time.
14) Lines 149, 169. "K" (kelvin) should be preferred to "°C".
15) Line 156. "Dispersivity" could be substituted with "Dispersion coefficient", as the term dispersivity is often used for the "dispersive length". The same term appears elsewhere, so if it is changed, it is necessary to check it throughout the whole manuscript. Notice that alpha appears earlier as one of the van Genuchten parameters: different symbols should be selected for these two quantities.
16) Line 170. I prefer the use of braces "{ }" to denote a set of variables, instead of square brackets.
17) Line 171. May be an adjective different from "complex" could be better, because, at a first reading I interpreted this as a complex number.
18) Line 175. Expression "besides better knowledge but due to the lack of any robust data" should be rephrased.
19) Line 185. Erase "," after "shown".
20) Line 194. I would use a n adjective different from "classical", may be "standard" or "common".
21) Line 217. "Per unit time" should be added after "melted", shouldn't it?
22) Line 223. "Note, that" can be erased.
23) Line 238. "Per unit time" should be added after "freeze", shouldn't it?
24) Line 253. Expression "of 13 - 90%" should be substituted with "between 13 % and 90 %".
25) Line 254. Expression "within the affects" should be corrected.
26) Line 268. IS "donating" the right word? May be, "denoting"?

27) Line 289. "A porous media" should be corrected: either singular (a porous medium) or plural (porous media).

28) Table 1, last line. It is not clear that "e" is used to denote scientific notation: $1.7 \times 10^{-3}$ and $1.7 \times 10^{-5}$ should be preferred. The same applies to lines from 480 to 482, 494 & 495,497, 530 to 532, and to the K values in Table 2.

29) Line 314. "10 hours" could be substituted with "10-hour-long".

30) Lines 319 & 320. Expression "of 0.1−0.5mm" should be substituted with "between 0.1 mm and 0.5 mm".

31) Line 324. Word "observerations" should be corrected.

32) Lines 325 & 327. The same format should be used for "first" and "second". May be "shallowest" and "deeper" or similar could be used.

33) Line 344. "Quantitative" or "qualitative"?

34) Line 381 & 382. Sentence "the effect... meteorological quantities" should be corrected.

35) Line 400. Expression "coincides similar" should be corrected.

36) Figure 4. "(c)" is missing in the figure caption. Expression "describe to the scenario" should be corrected.

37) Line 420. "between 20 to 70%" should be rephrased either as "between 20 % and 70 %" or "from 20 % to "70 %". Analogous corrections should be introduced in the following lines.

38) Table 2. The measurement units of h should be written more precisely as "$W/(m^2\,K)$".

39) Line 580. I would use "down to a depth of 30 cm" instead of "up to 30cm".

40) Throughout the whole paper, the space between values and measurement units is often missing.

41) The vertical axis of all the figures should be changed. At line 95, it is stated that the z-axis is assumed positive downwards. Therefore, it would be better to use positive numbers increasing downwards for the vertical axes of the figures. Moreover, the axis title is "depth of the snow column" and this is 0 at the top of the glacier or snowfield and increases downwards, so that it should be positive.

---

## Author Response (AR2)

**Response to Editor and Reviewers**

Both reviewers acknowledge the improvement of the manuscript after the revision and suggest "minor revision".
I warmly recommend the author to carefully revise the whole manuscript, and, in particular, to consider the following suggestions, based on the reviewers' comments.

**Reply: Thank you very much for your constructive feedback and giving me the possibility for further clarifying some statements made in the manuscript and highlight the research questions addressed.**

1) Improve the Introduction, especially the end of that section, to clarify what research questions are examined and how they are answered with the improved model.

**Reply: Thank you very much for pointing out that the formulation of the research questions addressed needed further clarification. The Introduction was restructured and the research motivation and addressed questions are now clearly stated at the end of the Introduction.**

2) Once the research question is posed in a clear way, a better description of the methods which are compared and how the comparison is conducted should be included quite easily in a subsection of Methods and/or Data sections. This is partly consistent with one of the comments by Referee #1 on the original version of the manuscript. Therefore, the comparison of the innovative modeling proposed here with previous or different approaches is not yet optimal. Also, Referee #2 recalls that a comparison with experimental or monitoring data is missing, but the comparison with the results of other models is nevertheless interesting.

**Reply: The methods of comparison with field data and of the used LTE model for comparison have been included in the manuscript in three subsections in the Methods section. Please note that a comparison with experimental data, as suggested by reviewer #2 was added to the manuscript in the previous revision!**

3) Referee #2 requires a sensitivity analysis with respect to ice density, because the use of different ice densities in the original and in the revised version seems to leave the results unchanged.

**Reply: The requested density analysis has been added to the results section (Fig. 4). Please note that the perceived change in ice density was only due to a typo in the original manuscript (use of two different values in different sections of the manuscript). Hence, in the previous revision the typo was corrected without changing the simulations. As shown in the requested analysis, an increased ice density does influence the results but on a very minor level not visible in the provided temperature-depth profiles.**

---

## Author Response (AR3)

Line numbers refer to the version with tracked changes.

1) Lines 81 & 82. Sentence should be rephrased: "it potentially affects" is not clearly related to the preceding part of the sentence (where the subject is plural, namely "LTNE effects").

**The sentence has been rephrased to: "This work investigates under which meteorological and hydraulic/thermal snow conditions LTNE can sustain and how LTNE potentially affects the thermo-hydraulic response of the snow pack to the rain."**

2) Lines 88 & 89. Sentence "Following studies varying meteorological and thermo-hydraulic parameters further reveal conditions promoting LTNE effects and address the uncertainty in model parameterization" should be rephrased.

**The sentence has been rephrased to: "Subsequent simulations varying meteorological and thermo-hydraulic parameters investigate which conditions promote LTNE and address the effect of uncertainty in model parameterization."**

3) Line 95. I prefer "matric potential" to "pressure head".

**Changed**

4) Line 98. It would be useful to recall the measurement units for $\alpha$, i.e., [m-1].

**Done. Also for n (-)**

5) Equation (3). I am afraid that this equation is incorrect. If I am not missing something, it does not correspond to equation (23) of van Genuchten (1980). Basically, $(\epsilon l,sat - \epsilon l,ref)1\text{-}m$ should be substituted with $(\epsilon l,sat - \epsilon l,ref)\times(1 - m)\text{-}1$. Am I wrong?

**You are right! Thank you very much for catching this error. As you can see in the Matlab function calcMoistCapac provided in the GitLab repository for the reviewers, this function was implemented correctly, so this is only a mistake in the manuscript.**

6) Lines 103, 104, 107, 108, 259. A space Is missing in "van Genuchten".

**Spelling was corrected throughout the manuscript.**

7) Line 108. A space should be added after "season".

**Added.**

8) Line 111. Word "been" is missing in "have also successfully used".

**Added.**

9) Equation (4). It is necessary to recall the measurement units for $\alpha$. In particular the equation could be rewritten as $\alpha$ = 7.3 m-1 × mm-1 × *d* + 1.9 m-1.

**The formula has been adjusted accordingly.**

10) Equation (5). Analogously to (4), a preferred format for this equation is n = -3.3 mm × *d* + 14.4.

**The formula has been adjusted accordingly. Please note it is** n = -3.3 mm$^{-1}$ × *d* + 14.4

11) Line 127. I think that "From rocks" should be substituted, possibly with "For saturated porous media".

**Rephrased as suggested.**

12) Equation (7). The definition of phi and phi0 is missing.

**Added.**

13) Line 142. Sentence "Throughout... respectively" should appear much early, when the indices for the three phases are used for the first time.

**The sentence has been moved following equation 2, when the subscript l is introduced for the first time.**

14) Lines 149, 169. "K" (kelvin) should be preferred to "°C".

**Done.**

15) Line 156. "Dispersivity" could be substituted with "Dispersion coefficient", as the term dispersivity is often used for the "dispersive length". The same term appears elsewhere, so if it is changed, it is necessary to check it throughout the whole manuscript. Notice that alpha appears earlier as one of the van Genuchten parameters: different symbols should be selected for these two quantities.

**The term Dispersivity was replaced by Dispersion coefficient throughout the manuscript (2 occurrences). The alpha was modified using a subscript T to distinguish it from the van Genuchten parameter.**

16) Line 170. I prefer the use of braces "{ }" to denote a set of variables, instead of square brackets.

**Square brackets changed to { } as suggested.**

17) Line 171. May be an adjective different from "complex" could be better, because, at a first reading I interpreted this as a complex number.

**The adjective has been removed completely as the meaning is explained in the following half-sentence anyway.**

18) Line 175. Expression "besides better knowledge but due to the lack of any robust data" should be rephrased.

**The sentence has been split into three sentences for a better reasoning.**

19) Line 185. Erase "," after "shown".

**Done**

20) Line 194. I would use a n adjective different from "classical", may be "standard" or "common".

**The adjective has been shared to common.**

21) Line 217. "Per unit time" should be added after "melted", shouldn't it?

**Yes, the clarification has been added.**

22) Line 223. "Note, that" can be erased.

**Done.**

23) Line 238. "Per unit time" should be added after "freeze", shouldn't it?

**Yes, the clarification has been added.**

24) Line 253. Expression "of 13 - 90%" should be substituted with "between 13 % and 90 %".

**Done.**

25) Line 254. Expression "within the affects" should be corrected.

**The sentence has been corrected to: Melting of snow and freezing of liquid water affect porosity, permeability, and heat transfer area of the snow.**

26) Line 268. IS "donating" the right word? May be, "denoting"?

**Yes, the spelling has been corrected to denoting.**

27) Line 289. "A porous media" should be corrected: either singular (a porous medium) or plural (porous media).

**The sentence has been corrected to "a porous medium".**

28) Table 1, last line. It is not clear that "e" is used to denote scientific notation: 1.7 x 10-3 and 1.7 x 10-5 should be preferred. The same applies to lines from 480 to 482, 494 & 495,497, 530 to 532, and to the K values in Table 2.

**The notation has been modified throughout the manuscript accordingly.**

29) Line 314. "10 hours" could be substituted with "10-hour-long".

**Done.**

30) Lines 319 & 320. Expression "of 0.1–0.5mm" should be substituted with "between 0.1 mm and 0.5 mm".

**Done.**

31) Line 324. Word "observerations" should be corrected.

**Done.**

32) Lines 325 & 327. The same format should be used for "first" and "second". May be "shallowest" and "deeper" or similar could be used.

**2ⁿᵈ has been corrected to second. Shallowest and deeper were added for clarification.**

33) Line 344. "Quantitative" or "qualitative"?

**"quantitative" but the sentences have been slightly rephrased for clarification.**

34) Line 381 & 382. Sentence "the effect… meteorological quantities" should be corrected.

**The sentence has been corrected.**

35) Line 400. Expression "coincides similar" should be corrected.

**The sentence has been corrected.**

36) Figure 4. "(c)" is missing in the figure caption. Expression "describe to the scenario" should be corrected.

**The (c) was added and the sentence was corrected.**

37) Line 420. "between 20 to 70%" should be rephrased either as "between 20 % and 70 %" or "from 20 % to "70 %". Analogous corrections should be introduced in the following lines.

**Done throughout the manuscript.**

38) Table 2. The measurement units of h should be written more precisely as "W/(m2 K)".

**Done as the unit notation has been fixed according to requirements throughout the manuscript W m-2 K-1**

39) Line 580. I would use "down to a depth of 30 cm" instead of "up to 30cm".

**Done**

40) Throughout the whole paper, the space between values and measurement units is often missing.

**The units have been corrected according to the Copernicus format.**

41) The vertical axis of all the figures should be changed. At line 95, it is stated that the z-axis is assumed positive downwards. Therefore, it would be better to use positive numbers increasing downwards for the vertical axes of the figures. Moreover, the axis title is "depth of the snow column" and this is 0 at the top of the glacier or snowfield and increases downwards, so that it should be positive.

**The figures have been revised accordingly.**

Moreover, I did not find a stability analysis of the numerical model, so that I wonder if the oscillations found in some cases (Figures 1 and 11) can be indeed explained as physical processes or if they are related to numerical instabilities.

**As stated in lines 282 – 284 (previous round of revisions, version with marked changes), the LTNE model requires an explicit time stepping, which is subjected to a stability criteria derived in Heinze & Hamidi (2017). This is the upper bound for the computational time step due to the comparably high difference in phase temperatures at the initial conditions and the time step is**

set constant throughout the simulation. Consistency of the numerical implementation was secured through variation of spatial and temporal resolution (lines 307-308). In addition, multiple parts of the code have been successfully validated with analytical solutions and compared to experimental data in past publications (such as Heinze & Hamidi, 2017; Heinze & Blöcher, 2019; Heinze, 2021).

Also, please note an important difference: The fluctuations shown in Fig. 1b (note the scale of the x-axis!) are numerically induced due to the corrector-predictor scheme applied to simulate the freezing/melting processes – but are no indications of instability. The variations seen in Fig. 11 are indeed interpreted as physical processes